



Atmospheric
Chemistry
and Physics

# Stratospherically induced circulation changes under the extreme conditions of the no-Montreal-Protocol scenario

**Franziska Zilker**[1,4]**, Timofei Sukhodolov**[2,3]**, Gabriel Chiodo**[1]**, Marina Friedel**[1]**, Tatiana Egorova**[2]**,
Eugene Rozanov**[2,3]**, Jan Sedlacek**[2]**, Svenja Seeber**[1]**, and Thomas Peter**[1]

[1]Institute for Atmospheric and Climate Science (IAC), ETH Zurich, Zurich, Switzerland
[2]Physikalisch-Meteorologisches Observatorium Davos/World Radiation Center, Davos, Switzerland
[3]Ozone Layer and Upper Atmosphere Research Laboratory, St. Petersburg State University,
St. Petersburg, Russia
[4]Department of Land Change Science, Swiss Federal Institute for Forest, Snow, and Landscape
Research (WSL), Birmensdorf, Switzerland

**Correspondence:** Franziska Zilker (franziska.zilker@wsl.ch)

**Abstract.** The Montreal Protocol and its amendments (MPA) have been a huge success in preserving the stratospheric ozone layer from being destroyed by unabated chlorofluorocarbon (CFC) emissions. The phaseout of CFCs has not only prevented serious impacts on our health and climate, but also avoided strong alterations of atmospheric circulation patterns. With the Earth system model SOCOLv4, we study the dynamical and climatic impacts of a scenario with unabated CFC emissions by 2100, disentangling radiative and chemical (ozone-mediated) effects of CFCs. In the stratosphere, chemical effects of CFCs (i.e., the resulting ozone loss) are the main drivers of circulation changes, weakening wintertime polar vortices and speeding up the Brewer–Dobson circulation. These dynamical impacts during wintertime are due to low-latitude ozone depletion and the resulting reduction in the Equator-to-pole temperature gradient. Westerly winds in the lower stratosphere strengthen, which is for the Southern Hemisphere (SH) similar to the effects of the Antarctic ozone hole over the second half of the 20th century. Furthermore, the winter and spring stratospheric wind variability increases in the SH, whereas it decreases in summer and fall. This seasonal variation in wind speed in the stratosphere has substantial implications for the major modes of variability in the tropospheric circulation in the scenario without the MPA (No-MPA). We find coherent changes in the troposphere, such as patterns that are reminiscent of negative Southern and Northern Annular modes (SAM and NAM) and North Atlantic Oscillation (NAO) anomalies during seasons with a weakened vortex (winter and spring); the opposite occurs during seasons with strengthened westerlies in the lower stratosphere and troposphere (summer). In the troposphere, radiative heating by CFCs prevails throughout the year, shifting the SAM into a positive phase and canceling out the ozone-induced effects on the NAO, whereas the North Pacific sector shows an increase in the meridional sea-level pressure gradient as both CFC heating and ozone-induced effects reinforce each other there. Furthermore, global warming is amplified by 1.9 K with regionally up to a 12 K increase over eastern Canada and the western Arctic. Our study sheds light on the adverse effects of a non-adherence to the MPA on the global atmospheric circulation, uncovering the roles of the underlying physical mechanisms. In so doing, our study emphasizes the importance of the MPA for Earth's climate to avoid regional amplifications of negative climate impacts.

## 1 Introduction

The emission of anthropogenic halogenated ozone-depleting substances (hODSs) has been predominantly responsible for stratospheric ozone depletion since the 1960s (Solomon, 1999). As a result, the Montreal Protocol and its amendments and adjustments (MPA) were ratified to phase out global ODS production and consumption (World Meteorological Organization, 2022). The MPA mitigated severe health impacts from harmful UV radiation and negative climate impacts (Barnes et al., 2019; Neale et al., 2021). It has been also recently shown that the MPA restrictions led to clear changes in vertical dynamical coupling between the stratosphere and the troposphere in the past decades with implications for the tropospheric circulation modes (Banerjee et al., 2020). Unlike the health and climate impacts, such circulation response to much stronger future effects of avoided chlorofluorocarbon (CFC) emissions has not been widely addressed.

As already known from historical ozone depletion, the influence from stratospheric circulation changes on the troposphere and surface can be considerable, especially in the Southern Hemisphere (SH) (Thompson and Solomon, 2002; Gillett and Thompson, 2003; Thompson et al., 2005). The Antarctic ozone hole caused polar stratospheric temperatures to decrease (through reduced absorption of solar radiation) by up to 12 K by the end of the 20th century (Randel et al., 2016; Calvo et al., 2017). As a consequence, the Equator-to-pole temperature gradient intensified, which in turn strengthened the polar vortex and caused a delay in its break-up in spring (e.g., Thompson and Solomon, 2002; Dennison et al., 2015). The large-scale SH tropospheric circulation responded to the stronger vortex with a poleward shift in the midlatitude (eddy-driven) jet stream, a positive trend in the Southern Annular Mode (SAM), an expanding Hadley cell and a subsequent expansion of the subtropical dry zone (Banerjee et al., 2020, and references therein). In the Northern Hemisphere (NH), the tropospheric and surface response to ozone depletion is less well established partly because long-term trends in Arctic ozone are much smaller than in the Antarctic (Karpechko et al., 2018; Eyring et al., 2021). However, model simulations (Calvo et al., 2015) and observations (Ivy et al., 2017) show that in individual years with strong ozone depletion in the Arctic, the Northern Annular Mode (NAM) shifts to a positive phase in spring, and ozone has been shown to play a sizable role in this link (Friedel et al., 2022). Arctic ozone can also affect tropospheric climate in a scenario with large $CO_2$ forcing (Chiodo and Polvani, 2019). Overall, the historical ozone-depletion period showed that CFCs have the potential to severely alter the stratospheric state via the ozone depletion they induce, and in turn they triggered sizable changes in the large-scale tropospheric circulation.

In the "world-avoided" scenario (a world without the restrictions of the MPA (No-MPA) and thus a continued unabated increase in CFCs throughout the 21st century), the coupling between the stratosphere and the troposphere would become stronger (Morgenstern et al., 2008). In the scope of this study we will focus on the changes in the polar vortices that have a direct effect on the tropospheric circulation, mostly regarding the dominant modes of tropospheric midlatitude variability, the NAM and SAM, and the North Atlantic Oscillation (NAO).

Models used in previous world-avoided studies are not fully interactive and have limited representation of tropospheric and surface processes (e.g., fixed tropospheric ozone in Newman et al., 2009, fixed sea surface temperatures and sea ice in Egorova et al., 2013, or prescribed chemistry in Goyal et al., 2019) and only briefly touched upon how the changes in the stratosphere affect the large-scale tropospheric circulation. Stronger polar vortices and a strengthening of the SAM with respect to the present day would be detectable by 2030 (Morgenstern et al., 2008). Newman et al. (2009) showed that the upper flank of the subtropical jet (30° N, 70 hPa) would significantly strengthen by 2065. Using a similar forcing, Egorova et al. (2013) reported a substantial weakening of the polar vortices and a shift in the NAM to a negative phase by 2100. This shift is consistent with what we know on how the stratosphere and troposphere are dynamically coupled (Kidston et al., 2015; Domeisen and Butler, 2020). A weakening of the stratospheric polar vortex leads to an equatorward shift in the tropospheric midlatitude jet and is associated with a negative phase of the NAO, NAM and SAM. The equatorward shift in the storm tracks goes along with anomalous surface temperature patterns. In the case of a negative NAO pattern, there is a warming over eastern Canada and cooling over northern Eurasia. In contrast, an intensification of the polar vortex leads to the opposite effect: a poleward shift in the tropospheric midlatitude jet and a positive SAM, NAM and NAO index, making the storm tracks stronger and more zonally oriented towards the pole (Kidston et al., 2015; Domeisen and Butler, 2020). In general, similar mechanisms may also be at work in the case of ozone depletion from unabated CFCs, but the sign and details of these mechanisms remain unclear in the context of world-avoided-scenario studies.

In addition to their role in destroying ozone, CFCs are important greenhouse gases (GHGs) that can thus directly affect surface temperature by trapping infrared radiation. Goyal et al. (2019) state that the MPA avoided around 1 K global warming by 2050, and Garcia et al. (2012) find a 2.5 K increase in global surface temperature by 2070, whereas Egorova et al. (2013) see only significant surface warming of up to 1 K over the South Pole and southern China and up to −2.5 K regional cooling in Eurasia and Argentina in 2100. In their most recent study, Egorova et al. (2023) report a surface warming of 2.5 K by 2100. However, to what degree CFCs have an impact on surface warming (via longwave trapping) or can be potentially offset by cooling resulting from ozone depletion is still controversially discussed (Velders et al., 2007; Goyal et al., 2019; Morgenstern et al., 2020; Chiodo

and Polvani, 2022; Morgenstern et al., 2021; Young et al., 2021). Taken together, the climatic impacts of unabated CFC emissions, in particular the role of direct (GHG) and ozone-mediated effects, remain poorly understood.

In this study, we complement Egorova et al. (2023), who examine the impacts of a No-MPA scenario by the end of the century with an Earth system model focusing on the ozone layer, surface air temperature, sea-ice cover and precipitation. We shed light on the mechanisms by investigating how ozone depletion (Sect. 3.1) changes the stratospheric circulation in a No-MPA scenario (Sect. 3.2) and how these changes manifest at the surface in the SH (Sect. 3.3) and in NH winter (Sect. 3.4), as well as how surface temperatures are affected (Sect. 3.5) by the end of 21st century with the fully interactive Earth system model SOCOLv4. We also aim to disentangle the impacts of stratospheric ozone depletion on the surface from the warming effect of abundant CFCs. Using such an extreme scenario allows for a very clear signal-to-noise ratio of the modeled response without the need for advanced statistical analysis.

## 2   Method

The Earth system model (ESM) SOCOLv4 (Sukhodolov et al., 2021a) was used to conduct the set of free-running experiments to distinguish between chemical (i.e., ozone-mediated) and radiative CFC contributions in the no-Montreal-Protocol scenario. SOCOLv4 consists of the interactively coupled Earth system model (MPIMET, Hamburg, Germany) (Mauritsen et al., 2019), the chemistry module MEZON (Egorova et al., 2003) and the sulfate aerosol microphysical module AER (Weisenstein et al., 1997; Feinberg et al., 2019), and thus, it includes most of the known atmospheric processes involved in the ozone net chemical production and transport, as well as its feedbacks from climate. Each experiment consists of three-member ensemble simulations with MPA (ref) and without MPA (noMPA) limitations, covering the period 1980–2100. The model boundary conditions mostly follow the recommendations of CMIP6 under the historical (1980–2014) and SSP2-4.5 (2015–2100) emission scenarios (Riahi et al., 2017). In the noMPA experiment, hODS surface mixing ratios have increased by $3\% \, \text{yr}^{-1}$ since 1987 (Velders et al., 2007) for regulated species. For unregulated species, we follow the recommendations of the World Meteorological Organization (2018) (see Egorova et al., 2023, for details). Throughout the study, we refer to hODSs as CFCs.

To distinguish between the direct greenhouse effect of CFCs and their chemical effects (i.e., ozone depletion), we have performed an additional model run, where increasing CFCs were active only chemically but not radiatively (the CFC fields of the ref run were prescribed in the radiation scheme) under SSP2-4.5. See Table 1 for further details.

In the results and discussion we mainly focus on the months of June, July and August (JJA), when the signal is most prominent, to discuss the mechanisms. Results for other seasons are shown in the Supplement. In all figures (if applicable) statistical significance is calculated similarly to a two-sided $t$ test at a 90 % confidence level following Gutiérrez et al. (2021) and all areas that are not statistically significant are stippled. Unless indicated differently, all figures show the ensemble mean.

## 3   Results

### 3.1   Ozone under the no-Montreal-Protocol scenario

First, we analyze the impact of a hypothetical No-MPA scenario on ozone and the subsequent variations in stratospheric temperature and zonal winds due to ozone changes. In the second part, we investigate how the (ozone-driven) stratospheric circulation changes, as well as the impact of CFCs, are linked to the large-scale tropospheric circulation and the surface.

In a scenario where the MPA is not in place, the abundant CFCs in the atmosphere severely reduce the global total ozone column to only 60 DU by the end of the 21st century in both noMPA experiments (Fig. 1a). Note that both noMPA scenarios are lying on top of each other, suggesting that radiative effects of CFCs alone do not affect the global ozone content. This severe ozone reduction is consistent with findings from, e.g., Garcia et al. (2012), who reported a collapse of the global ozone layer with ozone columns below 100 DU in a No-MPA scenario in the mid-21st century (see also, e.g., Goyal et al., 2019; Velders et al., 2007; Newman et al., 2009; Egorova et al., 2013, 2023; Young et al., 2021). Figure 1b–d show the zonal mean ozone of the world without a Montreal Protocol (noMPA, b) compared to the reference (c) and the difference (noMPA − ref) (d) by the end of the century for austral winter (JJA). The uncontrolled CFC emissions have increased the chlorine concentration by a factor of 20–80 compared to the reference by the end of the 21st century, which causes an ozone depletion by up to 90 % in the stratosphere, with the strongest reduction happening in the lower stratosphere. Gas-phase ozone destruction by chlorine is additionally accelerated by its heterogeneous activation on stratospheric aerosols and polar stratospheric clouds (PSCs), which also became much more widespread due to the temperature drop in the lower stratosphere (Fig. 2a, c). The cooling is especially prominent in the tropical lower stratosphere, where temperatures drop below the PSC Type 1 formation threshold of 195 K between 130 and 20 hPa (Fig. 2c) and $Cl_2$ (Fig. 2b, d) accumulates. This was also observed in Newman et al. (2009) and Garcia et al. (2012), and we also see an increase in PSCs Type 1 (nitric acid trihydrate, NAT, and supercooled ternary solution, STS) (Fig. 2a) in the tropics. PSCs Type 2 (ice crystals, when temperatures fall below 190 K) are

**Table 1.** List of the experiment simulation procedure (left) and investigated effects (right). The effect names are given in bold, and the line below indicates how they were deduced from the different experiments. All experiments were performed with the SSP2-4.5 scenario.

| Experiment | Simulation procedure | Effect |
|---|---|---|
| noMPA_CFCRadOff | – MPA not in place<br>– CFCs inactive for radiation<br>– One member 120 years<br>– Two members branched out after 2070 and simulated for 30 years | **CFC chemical effect**:<br>noMPA_CFCRadOff − ref<br><br>**CFC radiative effect**:<br>noMPA − noMPA_CFCRadOff |
| noMPA | – MPA not in place<br>– CFCs active for radiation<br>– One member 120 years<br>– Two members branched out after 2010 and simulated for 90 years | **Total effect**:<br>noMPA − ref |
| Reference (ref) | – MPA in place<br>– Three members 120 years | |

parameterized to only extend from 0–50° in each hemisphere in SOCOL (see also Steiner et al., 2021).

In the troposphere, ozone is depleted by up to 60 %, consistent with the documented impacts of ODS on tropospheric ozone via, e.g., changes in stratosphere–troposphere exchange (Banerjee et al., 2016; Shindell et al., 2013). At around 100 hPa in low latitudes, self-healing of the ozone layer occurs (Fig. 1d). With depleted ozone in high altitudes, UV radiation can penetrate further down and produce ozone there, as is also observed in Morgenstern et al. (2008), Egorova et al. (2013), and Egorova et al. (2023). In the No-MPA scenario by the end of the 21st century, ozone depletion is no longer subject to any season or restricted to the polar regions but is happening globally all year round (Fig. A1). We particularly want to highlight here the severe ozone reduction in the tropical lower stratosphere, which introduces new dynamical consequences compared to the past and present-day ozone-depletion effects (see next Sect. 3.2).

## 3.2 Stratospheric response

Here, we investigate the effect of the No-MPA scenario in JJA on zonal mean temperatures (Fig. 3c), zonal winds (Fig. 3f) and age of air (Fig. D1c) and go into the decomposition of the CFC chemical (Fig. 3a, d) and radiative effect (Fig. 3b, e) to investigate the processes and quantify their contributions to the total impact of No-MPA. Consistent with other world-avoided studies (e.g., Goyal et al., 2019; Garcia et al., 2012), the global ozone depletion by the end of the 21st century leads to a severe decrease in lower-stratospheric temperatures. Figure 3c shows the temperature response to the combined effect of the CFC chemical effect (ozone depletion), which mainly cools the stratosphere, and the CFC radiative effect, which warms the troposphere and parts of the stratosphere. Lower-stratospheric temperatures (100–20 hPa) drop by over 20 K and by over 30 K in the upper stratosphere

(3–0.7 hPa) as shown in Fig. 3a, c. This is coherent with the pattern of ozone anomalies induced by CFCs (Fig. 1d), indicating that the cooling is mostly due to reduced ozone absorption of shortwave solar radiation, as well as of longwave terrestrial radiation (Fig. 1a, d). The cooling is especially prominent in the tropics (Fig. 3a, c), where heterogeneous chlorine activation enhances ozone destruction (Fig. 2a, b). This severe cooling is seen throughout all seasons (Fig. B1). The area of reduced cooling between 20 and 3 hPa in the tropics and NH can be explained on the one hand by the maximum ozone concentration region at around 10 hPa (see Fig. 1b and c) and on the other hand by the increased absorption of infrared radiation at 9.6 µm, as missing ozone allows this radiation to penetrate higher up (Chipperfield and Pyle, 1988; Shine, 1986).

The drastic temperature changes in the stratosphere alter the lapse rate (Fig. C1) in the world-avoided scenario, lifting the tropopause in the tropics, which was also observed by Newman et al. (2009). The upward shift in the tropical tropopause (reaching 50 hPa) is almost entirely due to ozone depletion, with CFC radiation barely affecting it, as seen in Figs. 3b and C1. This effect is similar to the tropopause rise from well-mixed GHG (Santer et al., 2003; Meng et al., 2021). Above the tropopause, the stratospheric temperatures strongly increase up to the inflection point at 3 hPa, where they start to decrease again, suggesting that the stratopause drops to lower altitudes in the No-MPA experiments, shrinking the stratosphere compared to the reference.

Interestingly, the Antarctic stratosphere exhibits a warming of over 3 K at around 10 hPa (similar signal for Arctic stratosphere in Fig. B1d and f). Newman et al. (2009) explained it by an increased downwelling due to the Brewer–Dobson circulation (BDC) speedup. However, CO changes shown in Fig. D1d and f indicate reduced vertical transport from the mesosphere. CO can be interpreted as a dynamical tracer from its production region (CO$_2$ photolysis) in the

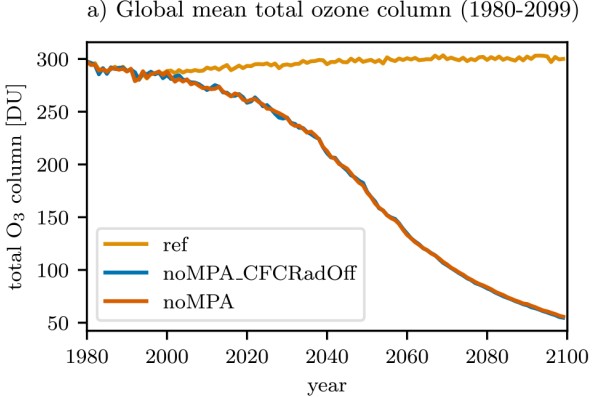

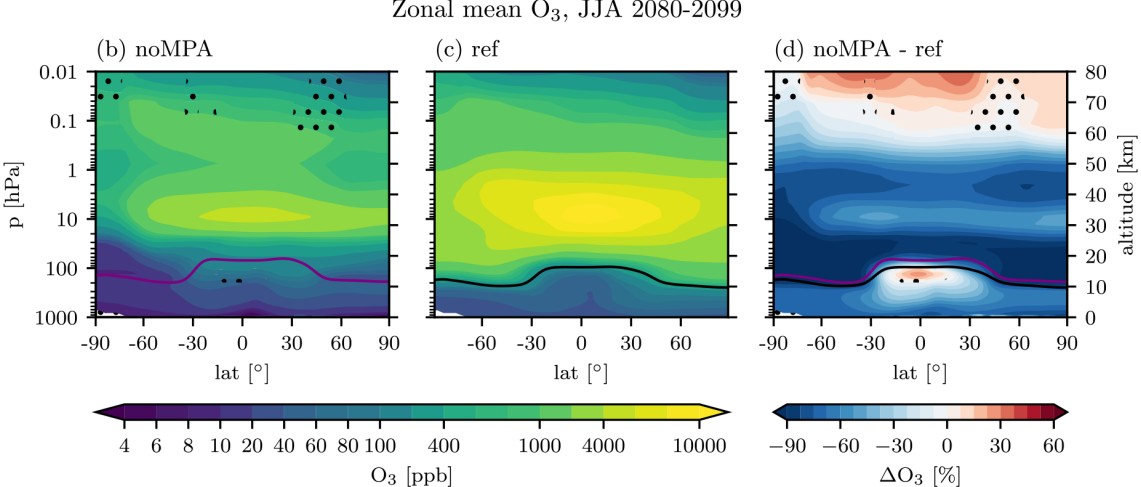

**Figure 1. (a)** Global mean total ozone column evolution from 1980–2099 for the reference, noMPA and noMPA_CFCRadOff scenarios. Bottom: zonal mean ozone for noMPA **(b)**, ref **(c)** and differences in percent of noMPA − ref **(d)** for JJA 2080–2099. The tropopause height is indicated in purple for the noMPA and in black for the MPA reference experiment. Stippling indicates not significant at a 90 % confidence level. Colorbar levels for panels **(b)** and **(c)** are evenly numbered in log spacing. Note that the color saturation for the difference is different for negative and positive values.

mesosphere (e.g., Solomon et al., 1985; Funke et al., 2009) and especially for the polar stratosphere (e.g., de Zafra and Muscari, 2004; McDonald and Smith, 2013). Funke et al. (2009) showed a very efficient CO descent in the mesosphere and stratosphere in the NH polar vortex during winter. However, when the vortex gets perturbed from a sudden stratospheric warming (SSW), this descent reduces. We argue that with the weaker vortex under No-MPA conditions, we have similar weak SH vortex conditions, which are reflected in the reduced polar CO in Fig. D1d and f due to the weaker mesospheric vertical transport. The warming at 10 hPa could then also be partly explained by the weaker vortex, allowing warmer air from the midlatitudes to be mixed into the polar stratosphere more easily. A similar warming has also been observed under ozone hole conditions by Haase et al. (2020) (see their Fig. 5) and Waugh et al. (2009) (see their Fig. 1).

CFCs by themselves (i.e., without considering their effects on ozone; Fig. 3b) strongly warm (by up to 5 K) the tropo-sphere, consistent with previous studies (Garcia et al., 2012; Goyal et al., 2019); we will examine this feature, along with surface temperature, in more detail in Sect. 3.5. The CFC-induced warming also extends into the lower stratosphere up to 20 hPa, which is consistent with the recent findings of Chiodo and Polvani (2022), indicating that this is a direct (radiative) effect without any influence of dynamical changes in this region. Upper-stratospheric warming at high latitudes in Fig. 3b most likely stems from the BDC speedup (see later in this section).

Next, we aim at understanding in more detail the dynamical impacts of a No-MPA scenario. In Fig. 3f the zonal wind response to the overall CFC effect is depicted. The wintertime polar vortex speed reduces substantially, whereas the subtropical jets (STJs) shift up and accelerate in both hemispheres. Furthermore, the BDC also speeds up, as the age of air gets younger in the entire stratosphere (Fig. D1c).

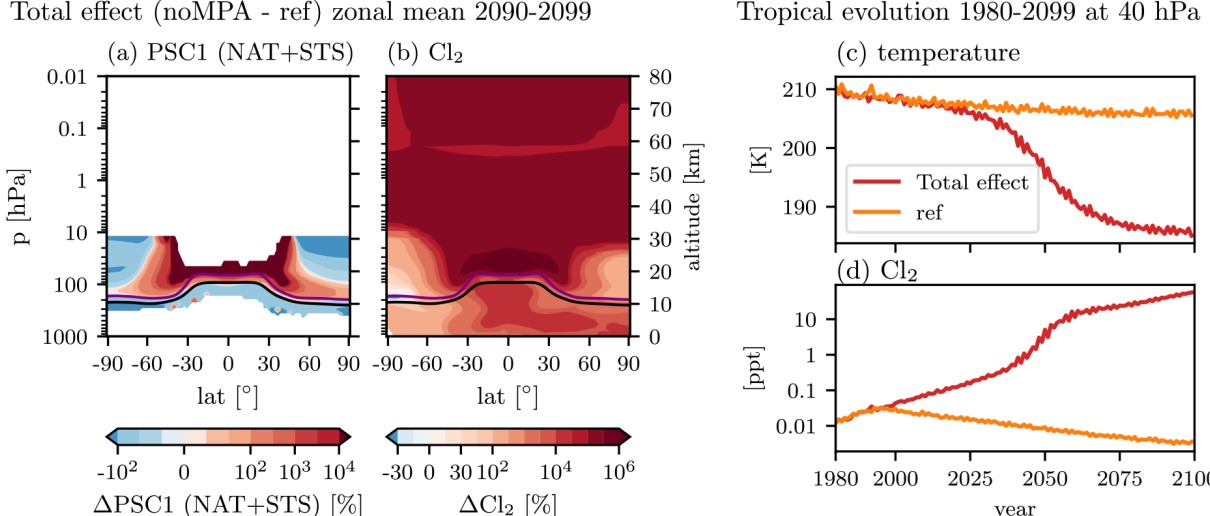

**Figure 2. (a, b)** The 2090–2099 annual mean zonal mean of PSC Type 1 (nitric acid trihydrate, NAT, and supercooled ternary solution, STS) **(a)** and $Cl_2$ **(b)** differences in percent (%) for the total effect (noMPA − ref). The tropopause height is indicated in purple for the noMPA and in black for the MPA reference experiment. Colorbar levels for panels **(a)** and **(b)** are linear around 0 and log spacing $< -10\%$ and $> 10\%$ for panel **(a)** and $> 30\%$ for panel **(b)**. **(c, d)** Evolution of tropical mean (23° N–23° S) temperature **(c)** and $Cl_2$ **(d)** anomalies at 40 hPa from 1980 to 2099 for the total effect (noMPA − ref). Note that panel **(d)** has a log $y$ axis.

As seen in Fig. 2c, the missing shortwave absorption from depleted ozone starts to reduce tropical temperatures in the lower stratosphere by 2030 and causes them to drop to 185 K by 2090. This severe cooling in the tropics reduces the Equator-to-pole temperature gradient (Fig. 3a). This reduction in the gradient is largest in the SH winter and starts to weaken the polar cap zonal wind at 10 hPa by 2040 (blue and red lines in Fig. 4). Consequently, the polar vortex slows down due to the severe cooling of the tropical lower stratosphere (around 50 hPa in Fig. 3a) from the CFC chemical effect, i.e., ozone depletion. By the end of the 21st century, the polar vortex in the SH has significantly slowed down by up to 25 m s$^{-1}$ at 10 hPa and 40 m s$^{-1}$ at 1 hPa (Fig. 3d). In the NH, we find a slowdown of the vortex by up to 15 m s$^{-1}$ at 10 hPa, although this signal is limited to individual seasons such as fall, winter and spring (Fig. E1). Egorova et al. (2013) observe a similar weakening of the polar vortices. For summer, we observe the opposite effect in both hemispheres. The stratospheric winds strengthen (Fig. E1d for SH and Fig. 3d for NH summer) due to stronger polar cooling than in winter, which increases the Equator-to-pole gradient again (Fig. B1d for SH and Fig. 3a for NH summer). Seasonally decomposing the vortex response for ozone depletion shows that the vortex reacts in a similar way to recent ozone-depletion trends in the summertime (strengthening of stratospheric westerlies), while it acts in the opposite direction in wintertime (weakening of stratospheric westerlies).

For the CFC radiative effect, we see that the polar vortices are slightly stronger by up to 10 m s$^{-1}$ in the SH (Fig. 3e) and NH during winter (Fig. E1e) compared to the CFC chemical effect. This enhancement originates from the CFC warming in the tropical troposphere and lower stratosphere. Therefore the Equator-to-pole temperature gradient in the lower stratosphere is larger when the CFC warming scenario is included. This is also seen in the SH polar cap wind evolution (purple line Fig. 4).

Additionally, we observe a strengthening of the upward flank of the subtropical jets near the tropopause and poleward shift (around 3° N and S) of the STJs throughout all seasons and scenarios (Figs. 3d–f and E1). Polar lower-stratospheric cooling during summertime further contributes to these dynamical changes, acting in the same way as Antarctic ozone hole conditions (e.g., Previdi and Polvani, 2014).

As a consequence of the weaker vortices and the stronger STJs, planetary waves can more efficiently propagate to the stratosphere. There they induce an acceleration of the BDC, leading to a decrease in age of air (AoA) in the global stratosphere (Fig. D1a–c), consistent with previous studies (Egorova et al., 2013; Newman et al., 2009; Morgenstern et al., 2008). Here, we find that this strengthening is almost entirely due to CFC-induced ozone depletion (Fig. D1a), similar to what occurred in the recent past (Abalos et al., 2019; Polvani et al., 2019). The strongest effect is on the shallow branch of the BDC, where the air gets younger by up to 0.8 years. The radiative heating by CFC further contributes to the speedup of the BDC (reduction of 0.3 years, mainly the deep branch; Fig. D1b), leading to a total AoA decline of 0.5 years in the deep branch and more than a year in the shallow branch (Fig. D1c).

In summary, the severe cooling from missing ozone has substantial implications for the stratospheric circulation, which, depending on the season, are the opposite to the his-

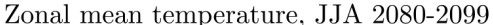

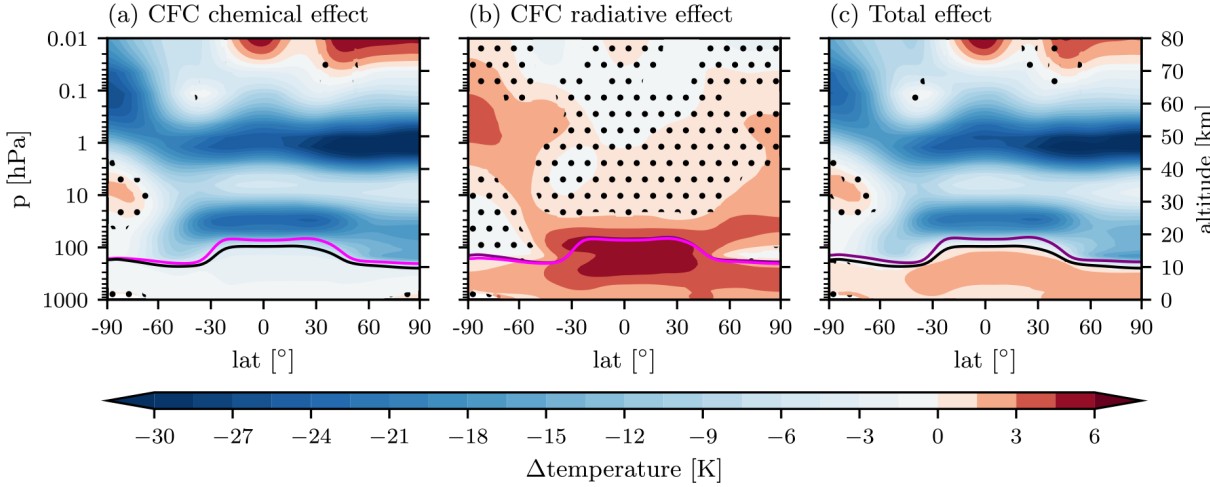

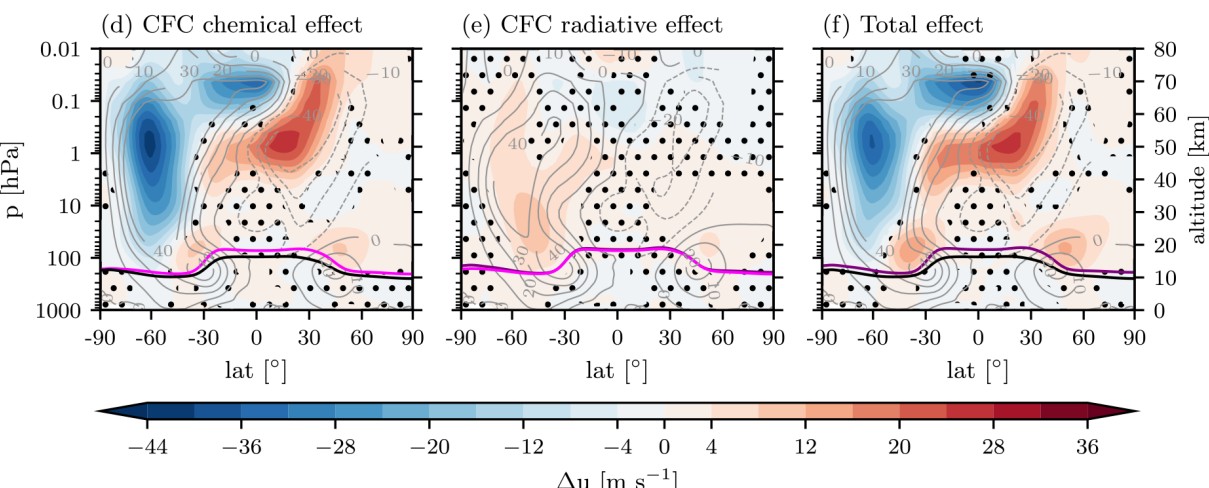

**Figure 3.** Zonal mean temperature differences **(a–c)** and zonal mean zonal wind differences **(d–f)** for JJA 2080–2099. The left column shows CFC chemical effect (noMPA_CFCRadOff − ref), the center column the CFC radiative effect (noMPA − noMPA_CFCRadOff), and the right column the total effect of CFC chemical and CFC radiative effects combined (noMPA − ref). The tropopause height is indicated in purple for the noMPA, in magenta for noMPA_CFCRadOff and in black for the reference experiment. Stippling indicates not significant at a 90 % confidence level. For zonal wind in the bottom row, the contour lines indicate the ref zonal wind profile. Note that the color saturation is different for negative and positive values.

torical ozone-depletion period (winter) or show the same sign (summer).

### 3.3 Implications for tropospheric SAM

To better understand the stratospheric implications of the No-MPA scenarios on the tropospheric variability modes, we focus on the SH polar vortex and its implications for tropospheric SAM. The SAM is a large-scale climate pattern in the SH with implications for temperature and precipitation. Figure 5a–c show the seasonal cycle of zonal wind changes between 40 and 70° S. As described in Sect. 3.2,

the wintertime polar vortex substantially slows down due to the CFC chemical effect (Fig. 5a, c). This weaker vortex in turn becomes more variable in winter and beginning of spring (Fig. 5d), as wave propagation into the stratosphere is facilitated, which increases the likelihood of sudden stratospheric warming (SSW). Morgenstern et al. (2022) showed that SOCOL is among the models that can generate SSWs in the SH. The weaker vortex in winter and spring manifests in the troposphere by pushing the tropospheric SAM to a more negative phase (Fig. 6a for winter, Fig. F1a for spring). This finding is a novelty to the current understanding of how ozone depletion can affect the SAM: in the extreme ozone-depletion

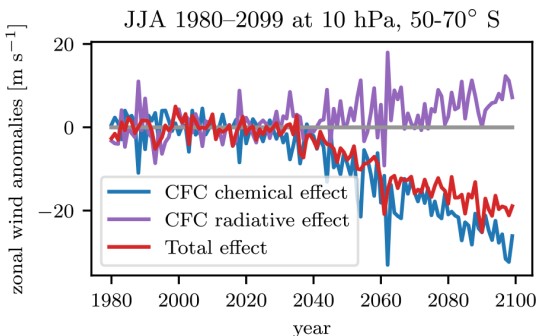

**Figure 4.** Zonal wind evolution at 10 hPa for 50–70° S for the CFC chemical effect (noMPA_CFCRadOff − ref), the CFC radiative effect (noMPA − noMPA_CFCRadOff), and the total effect of CFC chemical and CFC radiative effects combined (noMPA − ref).

scenario, the SAM shows a pronounced wintertime negative SAM phase and no longer strengthens in spring.

In contrast, winds in the stratosphere strengthen in summer and fall, effectively extending the lifetime of the strato-
5 spheric polar vortex (Sun et al., 2014) and shifting the tropospheric SAM to a more positive phase (Figs. 6d, F1d). This is consistent with our current understanding of how ozone depletion affects the stratospheric circulation and, in turn, how different vortex states affect the tropospheric circulation
(e.g., Domeisen and Butler, 2020).

For the CFC radiative effects, the opposite happens: the vortex strengthens in all seasons, causing a shift in the SAM to a more positive phase (SAM+) all year round (Figs. 6b, e, F1b, e). Additionally, the vortex variability
decreases (Fig. 5e). Combining both CFC effects shows (Figs. 6c, f, F1c, f) that the SAM+ response is dominated by the CFC radiative effect. It is only slightly reduced where the SAM is in a negative phase due to the CFC chemical effect (summer and spring) but reinforced where chemical and
radiative CFC effects contribute to the positive phase (winter and fall). This partial cancellation of effects (for summer and spring) is similar to what Morgenstern et al. (2014) found in terms of how ozone-mediated impacts of GHGs on the summer SAM offset the direct (radiative) effects of ODS and
GHGs.

Overall, the results show that the way ozone depletion affects the large-scale tropospheric circulation would have substantially changed in a future without the MPA as compared to how ozone depletion has affected the circulation in the
30 recent past. Until today, the CFC chemical effect has been presumably dominating the SAM response as it is driven by polar ozone depletion and recovery. In the No-MPA scenario, the polar ozone depletion gets nearly saturated towards the end of the 21st century. The polar vortex is then mostly af-
35 fected by the tropical ozone depletion, weakening the meridional temperature gradient and thus changing the sign of the stratospheric SAM and NAM anomalies. In addition we show that in the troposphere the CFC radiative effect be-

comes dominant, particularly towards the end of the century (see also Sect. 3.5). However, the overall SAM response is 40 still strongly modulated by the stratospheric changes, which strengthen or weaken the radiative effect, depending on the season.

Our findings are consistent with changes in wind at 500 hPa, which we use as a proxy for the eddy-driven 45 jet (Fig. F2). Most remarkably, the strongest response in the eddy-driven jet is seen in austral summer (December–February, DJF), when the jet strongly contracts poleward in the SH (Fig. F2c); this is due to the fact that during this season, chemical and radiative effects of CFCs act in the 50 same direction, in much the same way that GHGs and the ozone hole affected the westerly winds in the recent past (e.g., Previdi and Polvani, 2014). The poleward contraction and strengthening of the westerly winds, and in particular the shift to a more positive SAM phase, have wide repercussions 55 on regional weather regimes and precipitation patterns across the Southern Hemisphere (Gillett et al., 2006; Kidston et al., 2015; Brönnimann et al., 2017). Egorova et al. (2023) show in their Fig. 7b the 2090–2100 annual mean precipitation differences in the combined chemical and radiative CFC effects 60 (total effect). Precipitation increases over the Southern Ocean and Antarctica, while it decreases over South America and off the coasts of South Africa and southern Australia, consistent with the behavior of a positive SAM (Gillett et al., 2006). These regional precipitation changes are similar to what is 65 expected in a changing climate for the SH (Lee et al., 2023).

## 3.4  Tropospheric NAM response

The previous results show that the absence of the MPA and subsequent changes in ozone lead to drastic changes in the stratospheric circulation and also influence the tropospheric 70 circulation in the SH. Similar to the SH, large-scale climate variability in the NH can be described by the Northern Annular Mode (NAM) (Fig. 7 for winter, Fig. F3 for the other seasons). For the European and the North Atlantic sector, it is often referred to as the North Atlantic Oscillation (NAO) 75 (Eyring et al., 2021). The weakening of the polar vortex in the NH due to the CFC chemical effect, i.e., ozone depletion, is reflected in the decrease in the meridional near-surface pressure gradient in Fig. 7a. In boreal winter, we find a pressure increase at the NH pole and a decrease in the mid- 80 latitudes, which is reminiscent of a shift in the NAM to a more negative phase (NAM−). The sea-level pressure gradient forms a tripole-like pattern between the Atlantic and the Pacific. Since the pressure increase in the high latitudes is mainly in the Atlantic sector, we refer here to the NAM− as 85 NAO−. The NAO− pattern is also strongly reflected in the surface temperature response in Fig. 8b (DJF and March–May, MAM), with a warming over eastern Canada and a severe cooling over northern Eurasia.

For the radiative CFC effect, the NAM is zonally more 90 symmetric and shows the opposite signal all year round with

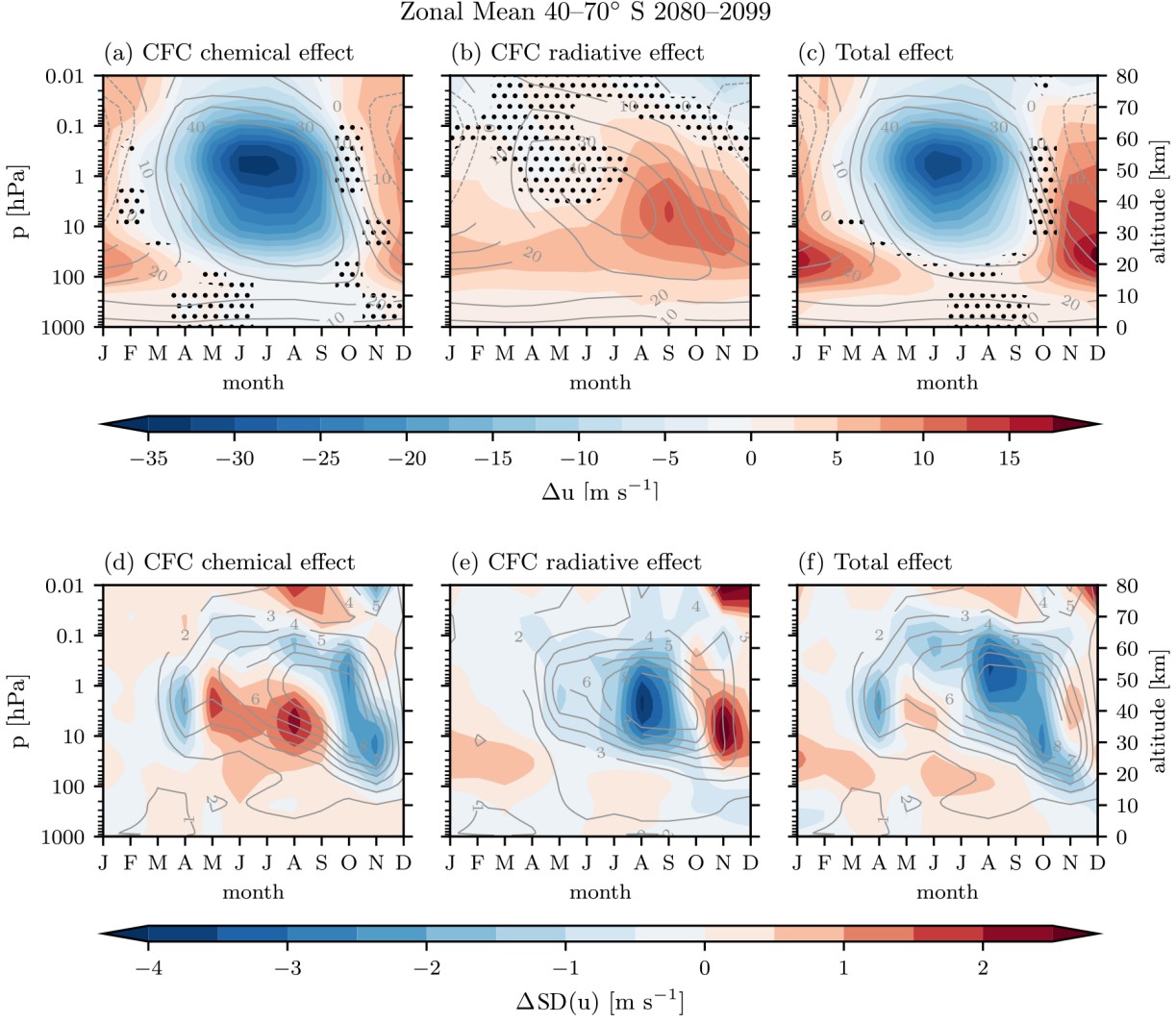

**Figure 5.** The 40–70° S zonal mean wind differences **(a–c)** and zonal mean standard deviation differences for zonal wind **(d–f)** for each month of 2080–2099. The left column shows CFC chemical effect (noMPA_CFCRadOff − ref), the center column the CFC radiative effect (noMPA − noMPA_CFCRadOff), and the right column the total effect of CFC chemical and CFC radiative effects combined (noMPA − ref). Stippling indicates not significant at a 90 % confidence level. The contour lines indicate the ref zonal wind profile **(a, b, c)** and the ref standard deviation **(d, e, f)**. Note that the color saturation is different for negative and positive values.

varying strength. It shifts to a more positive phase, i.e., the near-surface pressure gradient between the middle and high latitudes increases (Fig. 7b; for the other seasons Fig. F1 middle column). It is most likely that this response does not originate from the stratosphere, as stratospheric changes induced by CFCs are small (see Fig. 3b and e), but arises from the CFC-induced warming in the troposphere and in particular the upper tropical troposphere. This CFC GHG effect is similar to what is observed in a future changing climate (see, e.g., Ivanciu et al., 2022). The combined (total) CFC effects cancel each other out in winter in the NH Atlantic region (Fig. 7c); i.e., the NAO is unaffected by the collapse of the ozone layer. However, the Pacific sector shows a strengthening of the meridional pressure gradient as both chemical and radiative CFC effects reinforce each other there. The stronger pressure gradient over the North Pacific is associated with changes in regional precipitation patterns, such as enhanced precipitation over the North Pacific, Alaska, Canada and parts of the Arctic (Fig. 7b in Egorova et al., 2023). With climate change, Arctic latitudes are projected to receive more precipitation (Lee et al., 2023). Hence, the absence of the MPA would amplify the effects of climate change on the hydroclimate of these regions.

Overall under the extreme scenario of the No-MPA, we observe significant changes in the sea-level pressure anomalies both in the Atlantic and in the Pacific sector. However, the overall changes in the Atlantic sector cancel each other out due to the opposing signs of the individual effects. This

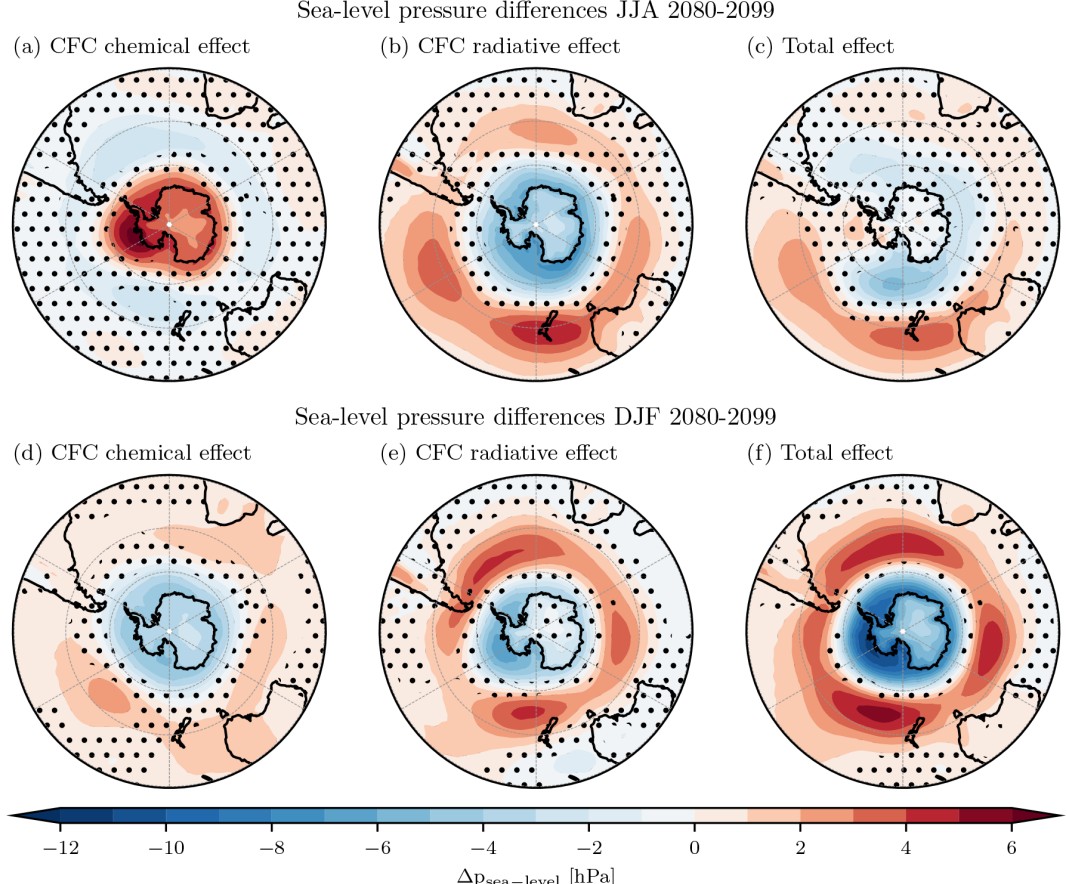

**Figure 6.** The 2080–2099 Antarctic winter **(a–c)** and summer **(d–f)** sea-level pressure differences. CFC chemical effect **(a, d)**, CFC radiative effect **(b, e)** and total effect **(c, f)**. Stippling indicates not significant at a 90 % confidence level. Note that the color saturation is different for negative and positive values.

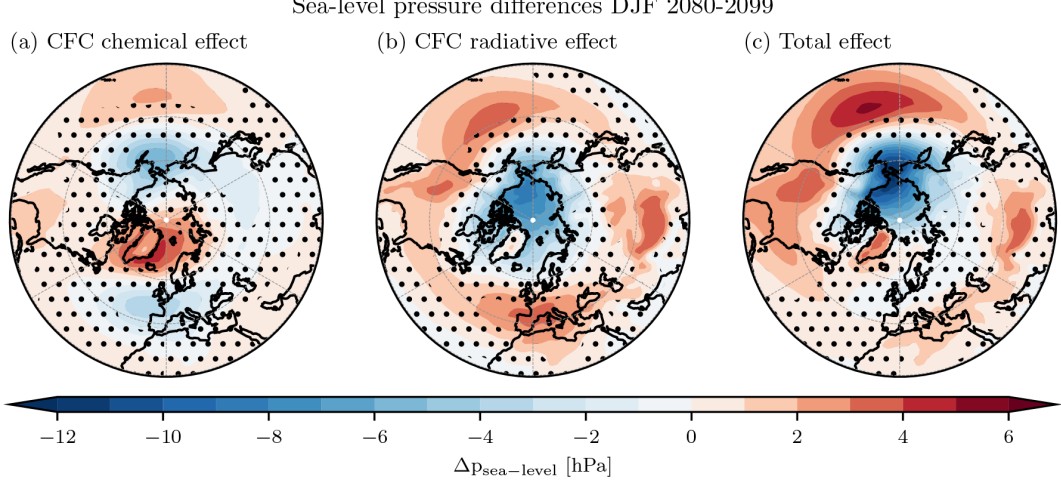

**Figure 7.** The 2080–2099 Arctic winter sea-level pressure differences. CFC chemical effect **(a)**, CFC radiative effect **(b)** and total effect **(c)**. Stippling indicates not significant at a 90 % confidence level. Note that the color saturation is different for negative and positive values.

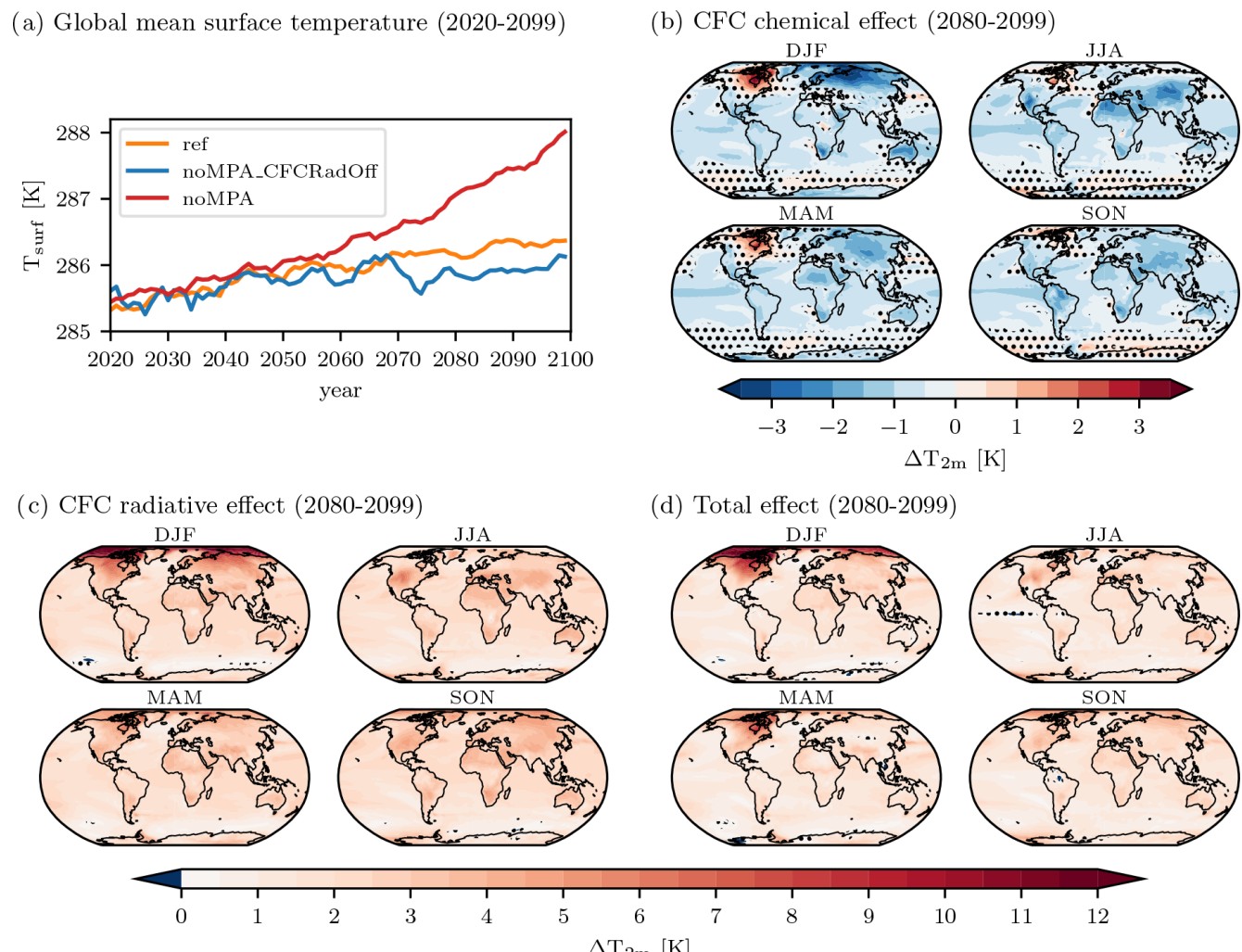

**Figure 8.** Surface temperatures: **(a)** global mean surface temperature evolution for the two noMPA scenarios and the reference from 2020–2099 and **(b–d)** global surface temperature for the CFC chemical effect, the CFC radiative effect and the total effect by the end of the century. Stippling indicates not significant at a 90 % confidence level.

is in contrast to today's knowledge of the effects of historical ozone-depletion trends in the Arctic, which are deemed unlikely to have induced any trends in the NAM (Karpechko et al., 2018). We also want to emphasize here that the forcing applied from this extreme No-MPA scenario exceeds by far any forcing from the historical ozone-depletion period.

### 3.5 Surface temperature response

As a consequence of the abundance of CFCs and their large greenhouse gas potential, the global surface temperature rises in the noMPA experiment by almost 3.5 K compared to 1980 and is around 1.9 K higher than the reference experiment by the end of the century (red line in Fig. 8a). This is similar to the warming by the end of the century obtained with the SSP5-8.5 scenario by the end of the century (see Fig. 4 in Egorova et al., 2023), as well as what Young et al. (2021)

show for their world-avoided study based on the RCP6.0 scenario. However, we are not taking into account the additional warming from the additional release of biospheric carbon in the No-MPA scenario as they did. Since the surface temperature changes are small in the 1980–2020 period (see also Fig. 4 in Egorova et al., 2023), we decided to only show it from 2020 to 2099 to zoom in on the period around 2050 when the temperature curves start to diverge due to the radiative CFC effect (see later this section). When the CFC warming is not considered (noMPA_CFCRadOff), the ozone depletion leads to a decrease in surface temperature by the end of the century (blue line in Fig. 8a) by 0.6 K compared to the reference (orange line in Fig. 8a). This temperature decrease is also depicted in Fig. 8b. For the boreal winter, we obtain the strongest cooling exceeding −3 K over northern Eurasia, whereas northern Canada experiences a warming of up to 3 K. This warming is most likely part of the dynami-

cal response and the resulting negative NAO phase due to the weakening of the stratospheric polar vortex discussed above. These regional temperature changes are due to, e.g., reduced advection of mild air over Eurasia (e.g., Hurrell, 1995; Visbeck et al., 2001). The CFC radiative effect increases the surface temperature by around 2.5 K globally, with the strongest signal being an over 12 K increase in the northern polar regions in DJF (Fig. 8c), which leads to a net warming of 1.9 K globally (Fig. 8d) compared to the reference. The most pronounced effect is seen over northern Canada where the warming from the ozone depletion adds to the CFC radiative effect, leading to an overall warming of over 13 K by the end of the 21st century compared to the reference. To put this warming into perspective, for the Arctic region, the IPCC AR6 (Lee et al., 2023) projects a warming of 10 K over the period from 1995–2014 to 2081–2100 under the highest-emission scenario SSP5-8.5. Hence, the warming in a mid-level-emission scenario (SSP2-4.5) without MPA would even surpass the warming in a high-emission scenario (SSP5-8.5) with the MPA being in place.

As seen from the temperature evolution in Fig. 8a, the CFC warming effect starts to overpower the cooling from ozone at around 2055. We consider this point in time to be the shift in regimes when the surface response to the No-MPA scenario is no longer dominated by the CFC chemical effect. The radiative effect of CFCs takes over and continues to modulate the surface climate, as was indicated by, e.g., Velders et al. (2007). Taken together, we find that the avoided warming due to the MPA is substantially modulated – at the regional scale – by the large-scale circulation changes induced by ozone and alterations in stratosphere–troposphere coupling. In addition, we find that globally, only a minor fraction (30 %) of the surface heating due to CFCs (via longwave trapping) is offset by the cooling due to the resulting ozone depletion (Goyal et al., 2019), consistent with recent work examining the radiative forcing (Chiodo and Polvani, 2022).

## 4 Conclusions

We conducted a set of experiments with the ESM SOCOLv4, where we investigated changes in the large-scale circulation of the stratosphere and troposphere under the extreme conditions of a no-Montreal-Protocol scenario by the end of the 21st century.

The key novelty over previous studies lies in our detailed separation of the effects induced by abundant CFCs: the chemical (i.e., ozone depletion) and radiative (i.e., global warming) properties of CFCs. To achieve this, we carried out experiments where CFCs were active and inactive for the radiation scheme. The main results of the CFC chemical effect are summarized as follows:

- Unabated CFC emissions deplete up to 90 % of ozone in the stratosphere by the end of the 21st century, severely

decreasing shortwave heating there and leading to a cooling of the global stratosphere by up to 30 K.

- The cooling is particularly pronounced in the tropical stratosphere, reducing the Equator-to-pole temperature gradient in both hemispheres and consequently also substantially weakening the winter polar vortices in both hemispheres.

- The weaker wintertime vortices shift the tropospheric SAM to a more negative phase in winter and spring, as well as the NAO (winter only). Additionally, the SH wintertime polar vortex variability decreases.

- In austral summer and beginning of fall, westerly winds in the SH stratosphere strengthen, causing a shift to a more positive SAM in the troposphere and a decrease in the wind variability.

- The global surface temperature decreases by 0.6 K with a regional warming of 3 K over northern Canada and cooling of $-3$ K over northern Eurasia. These regional patterns are largely modulated by the changes in the large-scale tropospheric circulation.

The CFC radiative effect counteracts the chemical effect of CFC-induced ozone depletion. Through their longwave absorptivity, CFCs strongly warm the troposphere (by up to 5 K) and the lower stratosphere. Further effects include the following:

- The tropical region is most affected by the CFC-induced tropospheric warming, which slightly increases the Equator-to-pole gradient, leading to slightly stronger wintertime vortices compared to the CFC chemical effect.

- The slightly stronger vortex, and thus decreased variability, together with the strong tropospheric warming of CFCs, shifts the tropospheric SAM to a more positive phase all year round and the NAM in winter only.

- The global surface temperature increases by 2.5 K with the strongest warming by up to 12 K over the Arctic regions.

Taken together, the CFC chemical effects largely shape the stratospheric temperature and circulation changes, whereas the CFC radiative effects are the dominant drivers of the large-scale tropospheric circulation and surface temperature changes. In the troposphere, the radiative effects of CFCs overcompensate for the changes resulting from ozone depletion (i.e., the CFC chemical effect). The combined CFC chemical and radiative effects are as follows:

- The BDC speeds up but with clearly distinct roles of chemical and radiative effects. The shallow branch is mostly affected by the CFC chemical effect, and the air becomes over a year younger, whereas the deep branch is mainly influenced by the CFC warming.

- Both effects cancel each other out for the NAO leaving it nearly unchanged under No-MPA conditions. In the North Pacific sector, both effects reinforce each other, increasing the meridional sea-level pressure gradient.

- The tropospheric SAM is more positive for austral summer and fall, when CFC chemical and radiative effects reinforce their positive phases, consistent with previous work on the ozone hole and its impacts on summertime circulation trends in the SH (World Meteorological Organization, 2018). The SAM+ signals weakens for winter and spring when both effects are in anti-phase.

- The global surface temperature increases by 1.9 K with the Arctic region being mostly affected in boreal winter (over 13 K warming) and spring, when both effects strengthen each other. The Antarctic region is fairly buffered and follows the mean global increase.

Overall, the MPA has not only prevented severe implications for our health, but also avoided substantial changes in our surface climate. Besides the well-known global warming effect of CFCs with subsequent tropospheric circulation changes, we showed that the dynamical changes in the stratosphere, caused by severe ozone depletion, would have also strongly affected the tropospheric variability modes, resulting in the regional amplification of adverse effects on surface climate. A further amplification of reduced precipitation over South America and increased precipitation over the Southern Ocean and North Pacific was avoided, as well as a further strengthening of the Arctic warming.

## Appendix A: Ozone

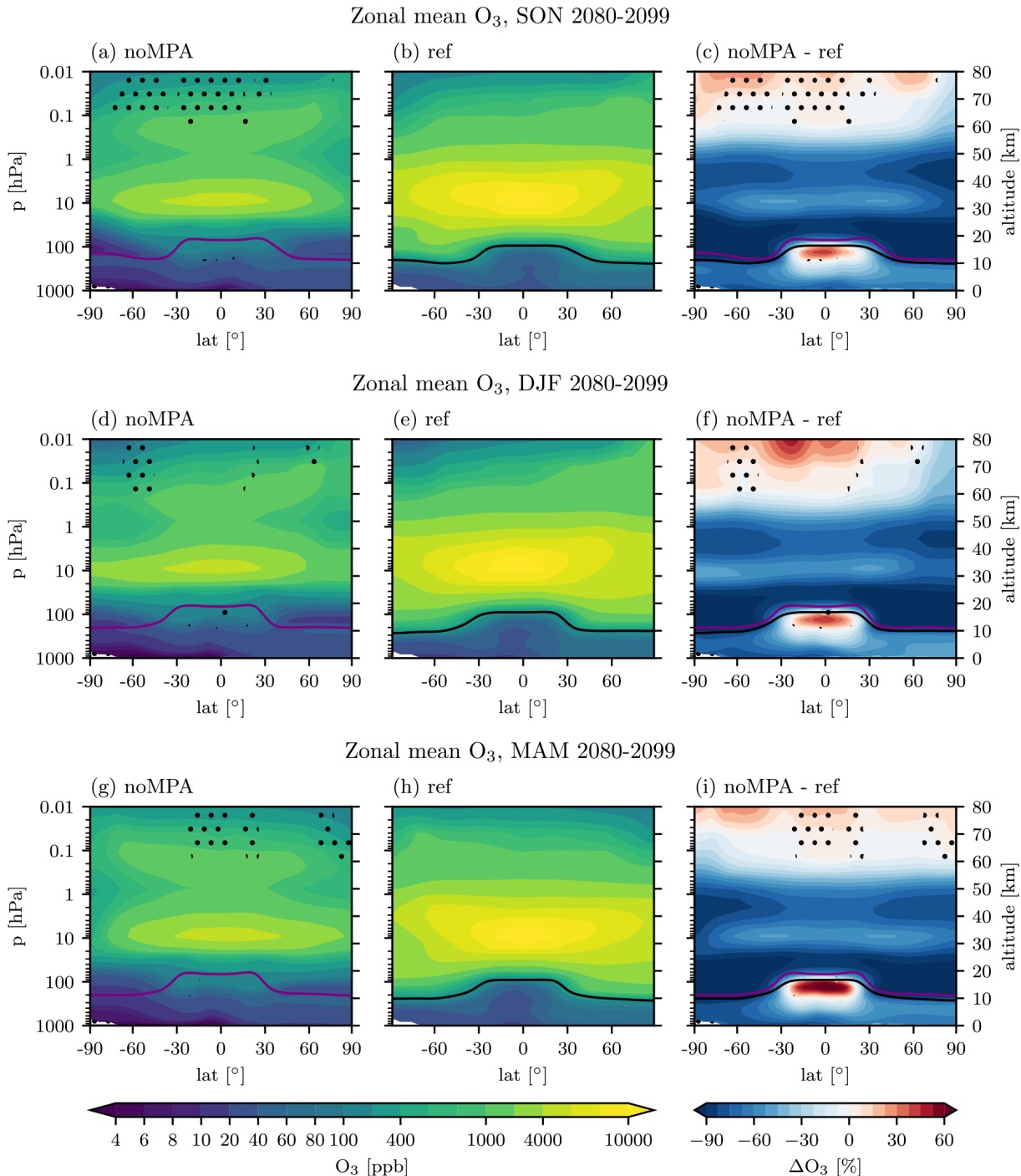

**Figure A1.** The 2080–2099 zonal mean ozone for noMPA **(a)**, ref **(b)** and differences in percent of noMPA − ref **(c)** for September–November (SON) **(a–c)**, DJF **(d–f)** and MAM **(g–i)** 2080–2099. The tropopause height is indicated in purple for the noMPA and in black for the MPA reference experiment. Stippling indicates not significant at a 90 % confidence level. Colorbar levels are evenly numbered in log spacing. Note that the color saturation for the differences is different for negative and positive values.

## Appendix B: Temperature

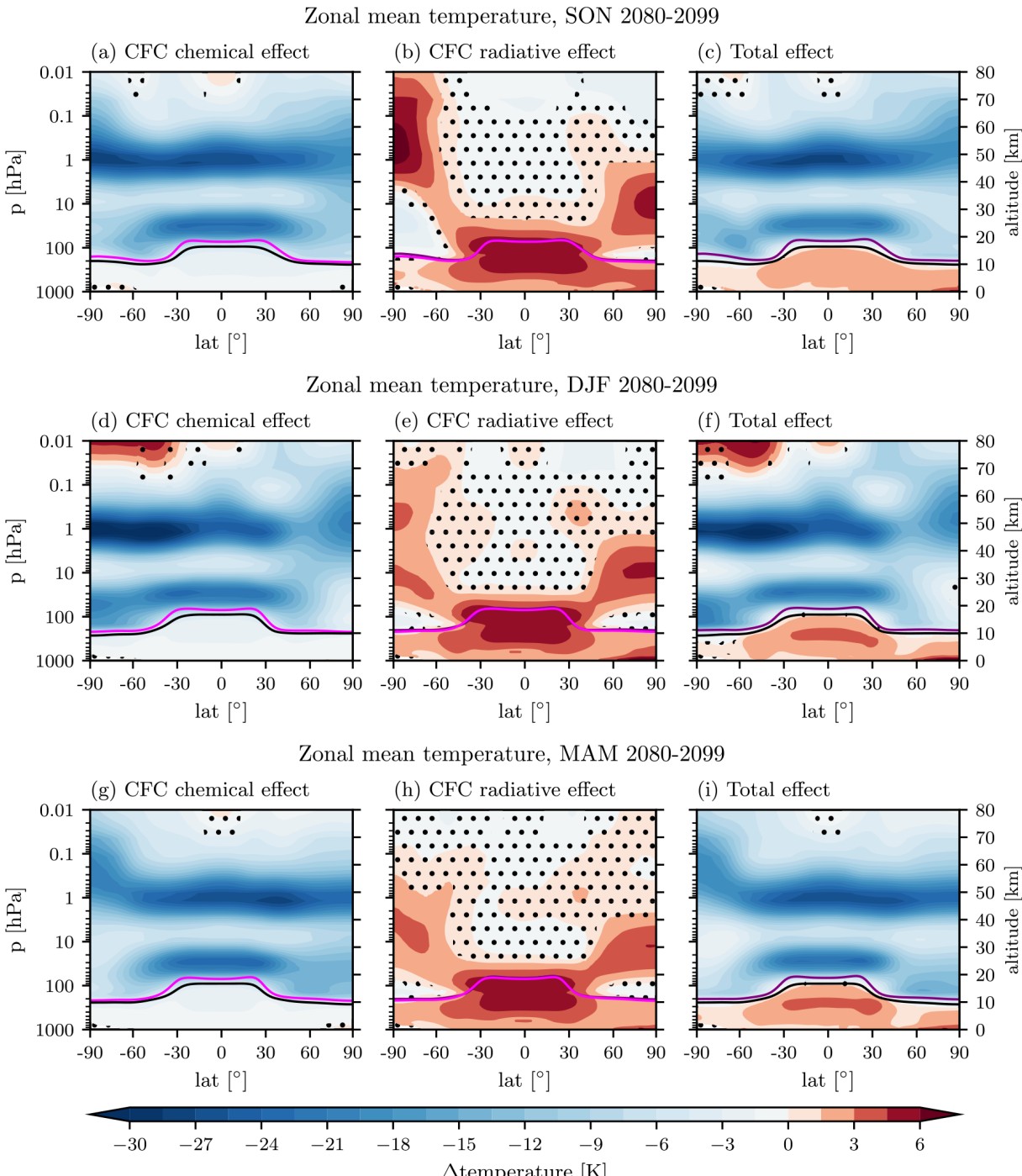

**Figure B1.** The 2080–2099 zonal mean temperature differences in percent for SON (**a–c**), DJF (**d–f**) and MAM (**g–i**). The left column shows the CFC chemical effect, the center column the CFC radiative effect, and the right column the total effect of CFC chemical and radiative effects combined. Stippling indicates not significant at a 90 % confidence level. The tropopause height is indicated in purple for the noMPA, in magenta for noMPA_CFCRadOff and in black for the reference experiment. Note that the color saturation is different for negative and positive values.

**Appendix C:  Temperature profile**

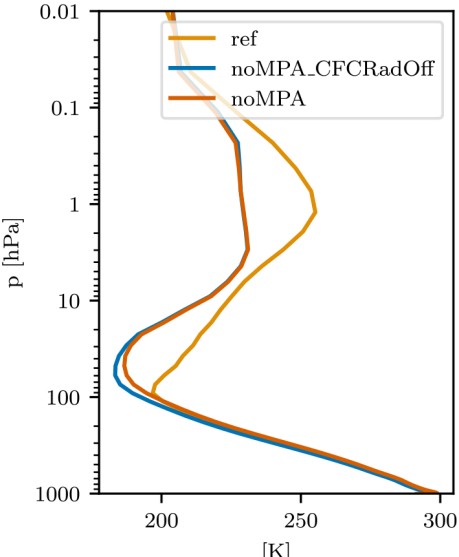

**Figure C1.** Tropical (30° N–S) zonal mean temperature profiles of noMPA, noMPA_CFCRadOff and ref in JJA 2080–2099.

## Appendix D: Brewer–Dobson circulation

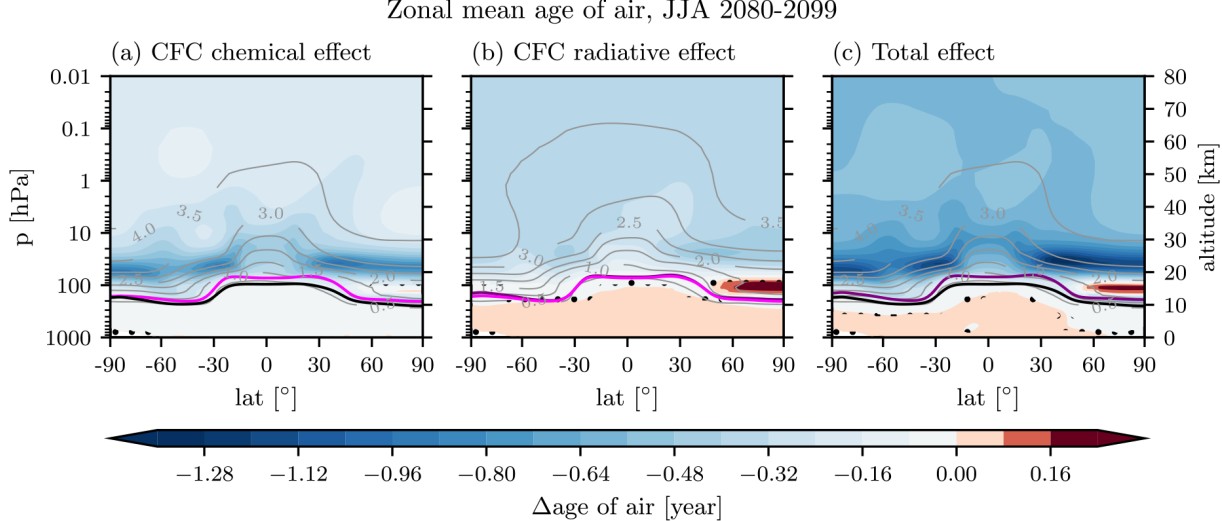

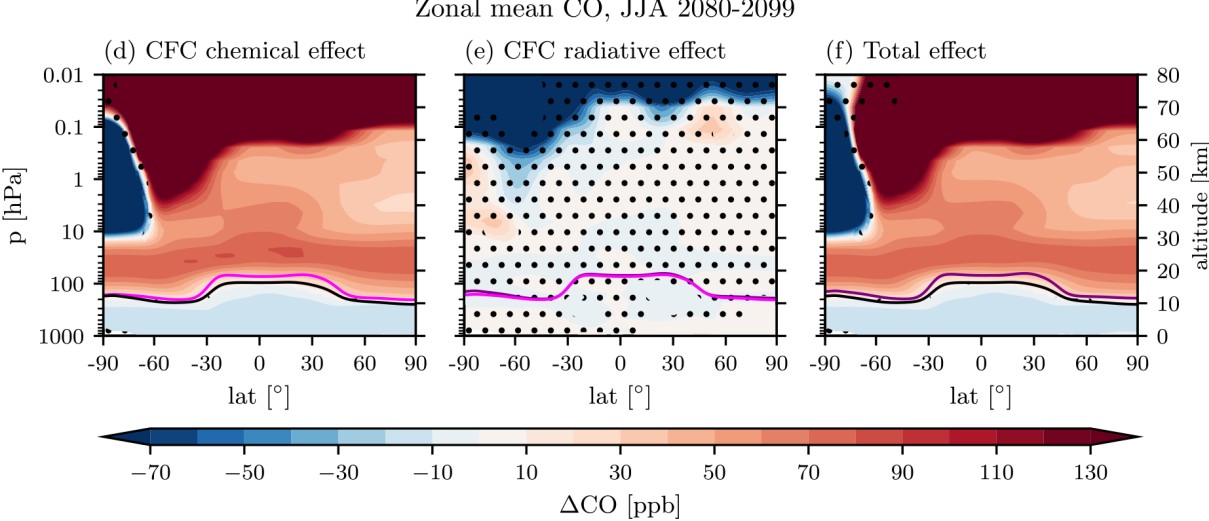

**Figure D1. (a–c)** Age of air for JJA (other seasons look very similar) 2080–2099 and **(d–f)** CO. The left column shows the CFC chemical effect, the center column the CFC radiative effect, and the right column the total effect of CFC chemical and radiative effects combined. Stippling indicates not significant at a 90 % confidence level. The tropopause height is indicated in purple for the noMPA, in magenta for noMPA_CFCRadOff and in black for the reference experiment. Note that the color saturation is different for negative and positive values.

**Appendix E: Zonal wind**

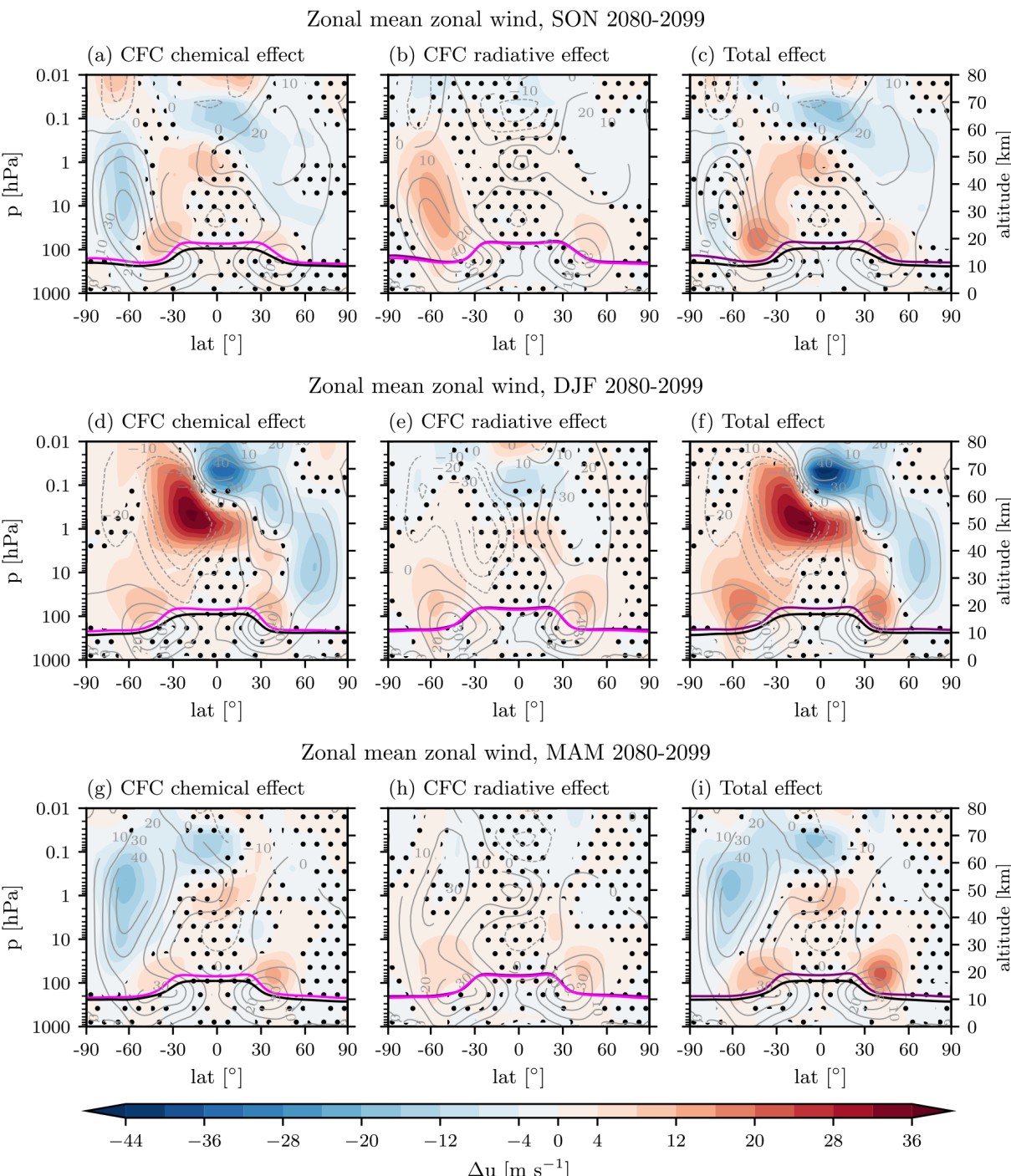

**Figure E1.** The 2080–2099 zonal mean zonal wind differences in percent for SON **(a–c)**, DJF **(d–f)** and MAM **(g–i)**. The left column shows the CFC chemical effect, the center column the CFC radiative effect, and the right column the total effect of CFC chemical and radiative effects combined. Stippling indicates not significant at a 90 % confidence level. The tropopause height is indicated in purple for the noMPA, in magenta for noMPA_CFCRadOff and in black for the reference experiment. The contour lines indicate the ref zonal wind profile. Note that the color saturation is different for negative and positive values.

## Appendix F: Surface

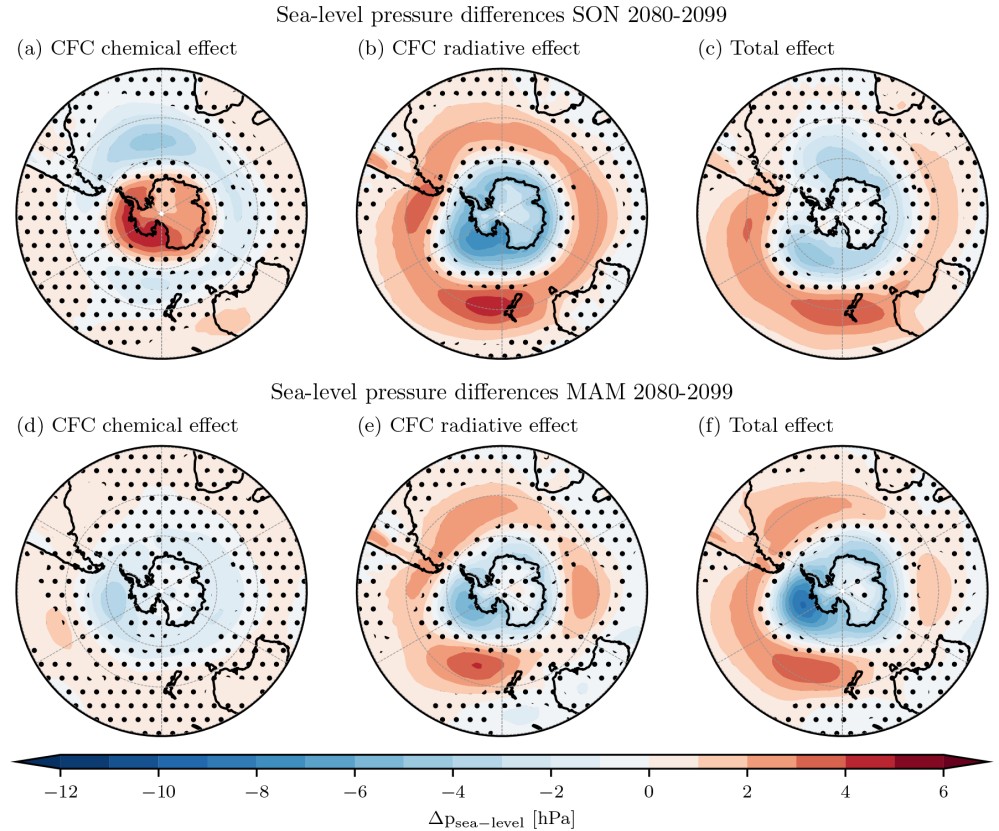

**Figure F1.** The 2080–2099 Antarctic spring and fall sea-level pressure differences. CFC chemical effect **(a, d)**, CFC radiative effect **(b, e)** and total effect **(c, f)**. Stippling indicates not significant at a 90 % confidence level. Note that the color saturation is different for negative and positive values.

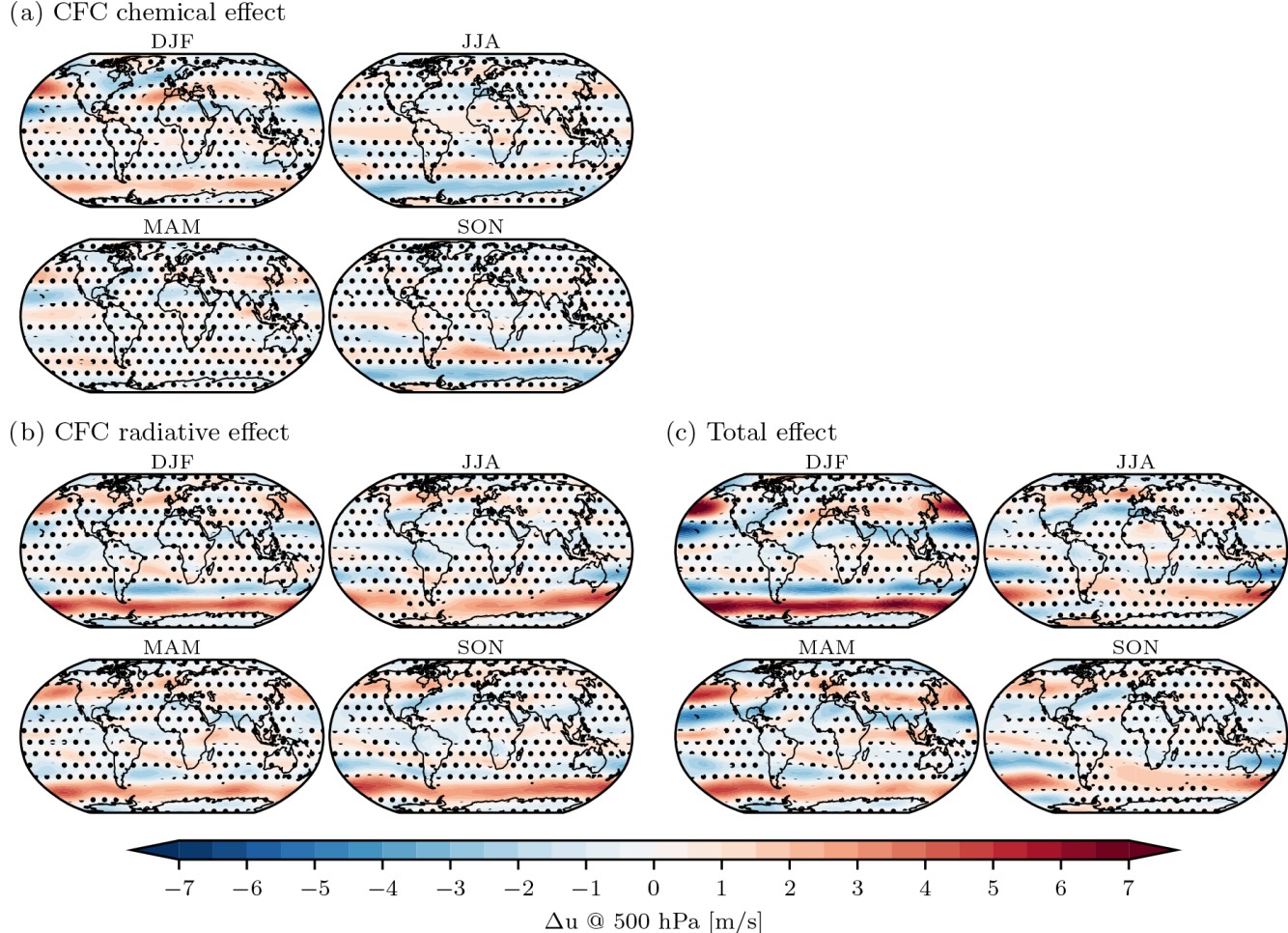

**Figure F2.** The 2080–2099 zonal wind differences at 500 hPa. CFC chemical effect **(a)**, CFC radiative effect **(b)** and total effect **(c)**. Stippling indicates not significant at a 90 % confidence level.

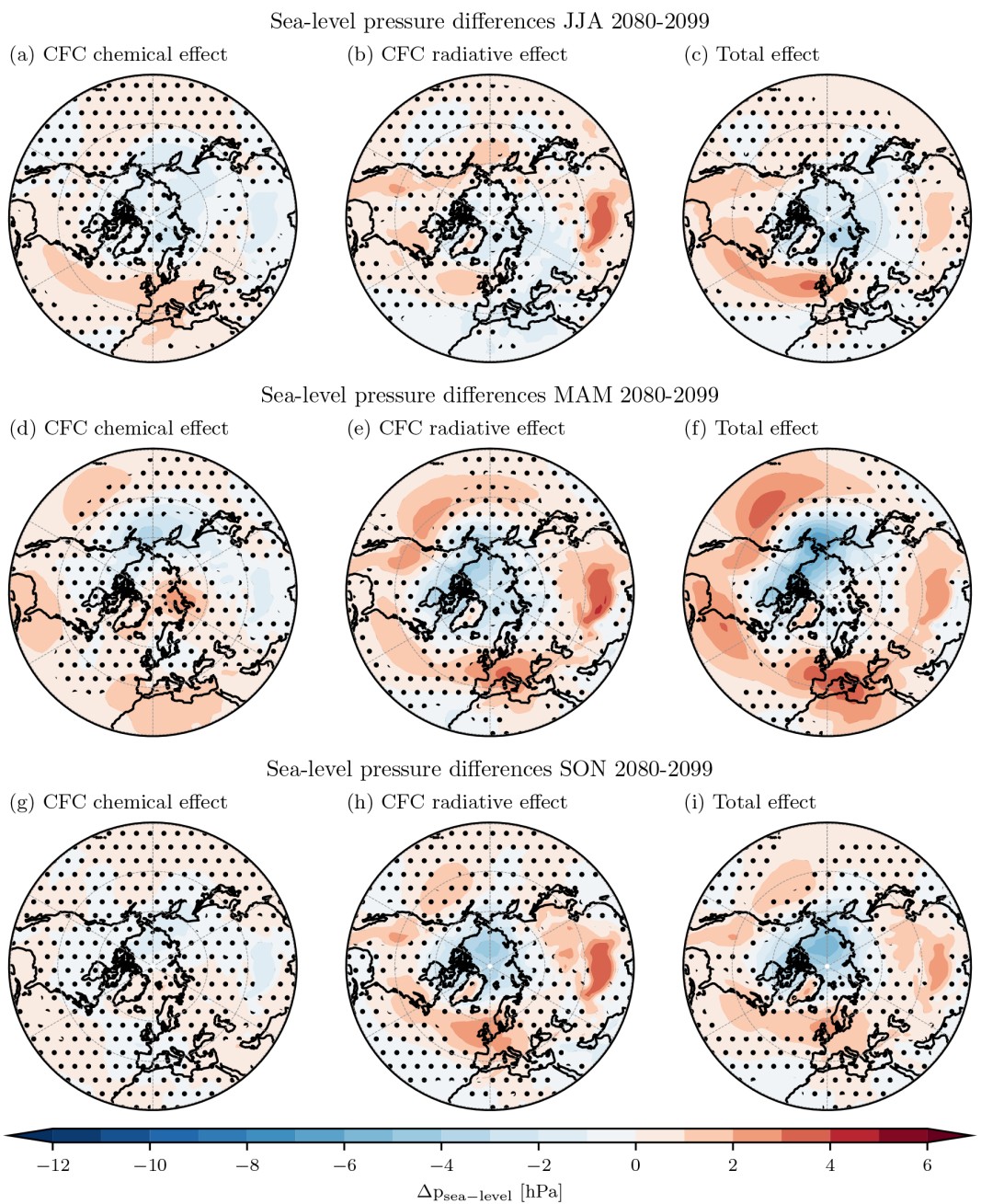

**Figure F3.** The 2080–2099 Arctic summer, spring and fall sea-level pressure differences. CFC chemical effect (**a, d, g**), CFC radiative effect (**b, e, h**) and total effect (**c, f, i**). Stippling indicates not significant at a 90 % confidence level. Note that the color saturation is different for negative and positive values.

**Code and data availability.** The code of SOCOLv4 is available in a general-purpose open repository on Zenodo at https://zenodo.org/record/4570622 (Sukhodolov et al., 2021b). Further information on SOCOLv4 can be found at https://doi.org/10.5194/gmd-14-5525-2021 (Sukhodolov et al., 2021a). The data were uploaded to a general-purpose open repository on Zenodo at https://zenodo.org/record/7234665#.Y1aP-UxBxaQ (Egorova et al., 2022) and it can also be provided by the corresponding authors upon request.

**Author contributions.** FZ performed the data analysis and visualization with support from JS and wrote the manuscript draft. TS and JS run the SOCOLv4 experiments. FZ, TS and GC analyzed and discussed the results. MF, SS, TP, TE, ER and JS participated in discussing the results and editing the manuscript.

**Competing interests.** The contact author has declared that none of the authors has any competing interests.

**Disclaimer.** Publisher's note: Copernicus Publications remains neutral with regard to jurisdictional claims in published maps and institutional affiliations.

**Acknowledgements.** Franziska Zilker thanks ETH Zürich and WSL for supporting this study. Tatiana Egorova, Timofei Sukhodolov, Jan Sedlacek and Eugene Rozanov thank the Swiss National Science Foundation for supporting this study through grant no. 200020-182239 project POLE (Past and future of the Ozone Layer Evolution). Jan Sedlacek and Tatiana Egorova received support from the Karbacher Fonds, Graubünden, Switzerland. Gabriel Chiodo, Marina Friedel and Franziska Zilker were supported by the Swiss National Science Foundation via the Ambizione grant no. PZ00P2_180043. Calculations were supported by a grant from the Swiss National Supercomputing Center (CSCS) under projects S1029 (ID 249), S1144 (ID 32696) and S1191 (ID 32797). The work was partially carried out at the St. Petersburg State University "Ozone Layer and Upper Atmosphere Research Laboratory" with the support of the Government of the Russian Federation (grant no. 075-15-2021-583). Data analysis and part of the model development were performed on the ETH Zürich cluster EULER.

**Financial support.** This research has been supported by the Schweizerischer Nationalfonds zur Förderung der Wissenschaftlichen Forschung (grant nos. 200020-182239 and PZ00P2_180043); the Karbacher Fonds, Graubünden, Switzerland; and the Government of the Russian Federation (grant no. 075-15-2021-583).

**Review statement.** This paper was edited by Peter Haynes and reviewed by two anonymous referees.

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
