# Peer review of "Stratospherically induced circulation changes under the extreme conditions of the No-Montreal-Protocol scenario"

_EGUsphere, 2023_

## Author Response (AR1)

**Authors' Final Response**

Franziska Zilker[1,4], Timofei Sukhodolov[2], Gabriel Chiodo[1], Marina Friedel[1], Tatiana Egorova[2], Eugene Rozanov[2,3], Jan Sedlacek[2], Svenja Seeber[1], and Thomas Peter[1]

[1]Institute for Atmospheric and Climate Science (IAC), ETH, Zurich, Switzerland
[2]Physikalisch-Meteorologisches Observatorium Davos/World Radiation Center, Davos, Switzerland
[3]St. Petersburg State University, St. Petersburg, Russia
[4]Swiss Federal Research Institute WSL, 8903 Birmensdorf, Switzerland

**Correspondence:** Franziska Zilker (franziska.zilker@wsl.ch)

We thank the editor and reviewers for their insightful comments and suggestions to our manuscript. We responded to them in detail below. The reviewers' comments are given in black and our answers are indicated in blue. We also explained minor corrections introduced by ourselves. The line numbers indicated in the response refer to the revised manuscript. The author's track-changes file is attached with all changes highlighted (generated with latexdiff).

**Answer to RC1**

**Summary:**

The authors conduct sensitivity simulations using the SOCOLv4 chemistry-climate model of a no-MP scenario involving continuing accelerating increases of atmospheric abundances of ozone-depleting substances. The unique aspect of this paper is that they decompose the modelled response into chemical (caused by ozone responses) and radiative (due to LW absorption by CFCs) effects. They show, for this extreme scenario, some partial cancellation of the effects of these two mechanisms. The results are consistent with and expand what had previously been known thanks to earlier studies of the same subject. I recommend publication of the study in ACP subject to addressing my comments, listed below.

**SH**

I think there is an interesting difference between some results reported here and what have been understood to be the present dynamical consequences of ozone depletion. Canonical wisdom is that ozone depletion causes a strengthening of the SAM due to cooling of the polar vortex; this influences tropospheric circulation in late spring and summer. Here the authors show that in the extreme ozone-depletion scenario of no Montreal Protocol, this is no longer the case. As ozone depletion becomes global, the cooling difference associated with this between low latitudes and the South Pole reduces, causing a weakening of the SAM. This had been new to me. The net strengthening of the SAM is then not mainly driven by ozone depletion but by the direct radiative heating due to CFCs. More could be made of this, and an explanation could be added that this is actually different from what is seen in the present situation when ozone depletion is not near saturation. This is my understanding; please feel free to agree or disagree with this.

Thank you for your concise summary, we completely agree that the No-MPA scenario leads to a shift in regimes that drive the tropospheric SAM, even though the overall result (SAM+) is the same as in the past and present ozone depletion period. We highlighted in the text the novelties of our results for both the stratosphere and the troposphere as well as their seasonal dependence and clarified the differences to the current status for the SH. See lines 132-134, 190-191 for the stratosphere and lines 218-223, 232-240 for the troposphere.

**NH**

For the Arctic, you discuss the "North Atlantic Oscillation". Please consider replacing this with the more generic "Northern Annular Mode". The NAO, in my perception, is just a regional expression of the NAM defined by the pressure difference between Iceland and the Azores. I'd prefer to use the "NAM" terminology here, in analogy with the SAM that you discuss throughout the paper. Also more substantially, under present conditions relatively moderate ozone depletion is not known to drive a trend of the NAM (which generally remains unexplained, see Ch3 of IPCC AR6) but there are associations between extreme ozone states and NAM anomalies, as you correctly state. Under the much more extreme ozone depletion considered here, you find some significant SLP anomalies. Perhaps more could be made of the fact that these effects are significant here but are not significant in the real world?

Thank you for your remark. We follow your line of argumentation and changed the wording to NAM where appropriate. Since we observe a more NAO-like pattern for the chemical CFC effect, we keep the term there, but putting it into perspective with NAM. As the pattern is more NAM-like for the radiative effect, we adapted the term there. Furthermore we highlight that the effects of long term ozone changes are significant in the No-MPA scenario in contrast to historical Arctic ozone depletion as you suggested (lines 13, 16-17, 41, the whole section 3.4 on tropospheric NAM and the conclusion line 318).

**Minor comments:**

Title: I'd drop the word "tropospheric" in the title as the paper equally deals with stratospheric circulation changes.

Thank you, that's a good point, we removed the "tropospheric"

L8: Usually during summer there is no "polar vortex" over Antarctica – it breaks down in spring. You probably want to state first that the polar vortex now persists year-round in the no-MP scenario (if that's the case).

Yes you are right, thank you. We revised the formulation in the the text. We wanted to say that lower stratospheric westerlies, such as the sub-tropical jet, strengthen throughout all seasons (lines 8-9).

L12: Insert "anomalies" before "during".

Thank you, we inserted it in the text.

L49: The "studies" are very interactive, just their models are not. Suggest to rephrase this, such as "Models used in previous studies are not fully interactive...".

Thank you for the suggestion, we inserted it in the text (line 52).

L69: "gases"

Thank you, we corrected it in the text (line 68).

L72-73: Young et al. (2021) is the most recent paper to deal with this; they add another 0.8 degrees of warming under the No-MP protocol, due to biospheric release of carbon under ozone depletion. Also https://www.nature.com/articles/s41598-019-48625-z could be discussed somewhere.

Thank you for mentioning this article. We added it in line 75 and discussed it briefly in the surface temperature section in lines 275-277.

L140: How do you know it's shortwave absorption? Ozone also absorbs outgoing LW radiation.

Yes, indeed. We added that the cooling is coming from missing shortwave absorption as well as the missing longwave terrestrial absorption. However, we do not have any model output to further distinguish their contributions and therefore mention that they both contribute (line 144-145).

L153: Morgenstern et al. (2018) find a similar feature (an increase in ozone in the polar middle stratosphere due to increasing chlorine in spring), across several CCMI models. It seems to be part of the dynamical response to ozone depletion.

Thank you for making us aware of their finding. Quoting their interpretation for Fig. 7 here: "In January, in what is likely a dynamical feedback, there is an increase in ozone (for an increase in ODSs) between about 50 and 10 hPa." We are not sure if we can compare our results and theirs, since we are mainly focusing on JJA here. Additionally, in their Fig. 7, the panel for July does not show any increase in ozone for an increase in chlorine. Since they do not further elaborate on potential reasons we refrain from indicating them in our text.

Figure 6: I'm impressed at how different the direct and indirect effects of CFCs on SLP are. This is to my understanding quite different again from the historical and present-day situation where I'm sure the direct effect is smaller than the indirect one. Perhaps you can comment on this. Also here a nod to Velders et al. (PNAS, 2007) might be in order who first stipulated that CFCs would be rivalling CO2 as the leading cause of global warming under a no-MP scenario.

To your question on the evolution of the chemical (indirect) and radiative (direct) effects of CFCs compared to the historical and present-day situation: We also agree that the indirect effects are stronger in the past and present-day ozone depletion period. However, we did not elaborate it in detail, but from the surface temperature evolution in Fig. 8a, we see that the CFC radiative effect becomes dominant from the 2050s on. There the temperature curves start to diverge. Therefore, we think that until this time, the chemical CFC effect dominates and then gets overpowered by the radiative CFC effect. We clarified this in the text (lines 218-223, 232-240) and also acknowledged Velders et al. (2007) in the section on surface temperature 3.5 (lines 291-292).

L251: Replace "largely" with "substantially". Also line 265.

Thank you for the suggestion, we added it to the text.

L275: Replace "mostly" with "most".

Thank you, we replaced it

Figure A1: The labels for (b), (e), (h) should be "ref" not "noMPA", and for (c), (f), and (i) probably "(noMPA – ref)/ref"

Yes, thank you. We adapted the labels accordingly.

[Figure]

**Figure A1.** 2080–2099 zonal mean ozone noMPA (a), ref (b) and differences in % of noMPA-ref (c) for SON (top), DJF (middle) and MAM (bottom) 2080-2099. The tropopause height is indicated in purple for the noMPA and in black for the MPA reference experiment. Stippling indicates not significant at a 90 % confidence level. Colorbar levels are evenly numbered in log spacing.

Figure B1: The usage of CO as a diagnostic probably needs more explanation (or dropping) as CO is not elaborated in the text and is a somewhat separate story.

We make use of CO as a dynamical tracer for the upper stratosphere and mesosphere (Solomon et al., 1985; Funke et al., 2009) especially for the polar stratosphere (de Zafra and Muscari, 2004; McDonald and Smith, 2013) to better understand the circulation patterns inside and around the vortex and to explain the warming at 10 hPa. The reduction of CO inside the polar vortex confirms on the one hand that the polar vortex becomes significantly weaker during No-MPA conditions. On the other hand the warming at 10 hPa most is likely coming from increased mixing of air from them mid-latitudes into the polar stratosphere due to the weaker polar vortex and not due to increased down-welling from the BDC as Newman et al. (2009) suggested. We tried to make this more clear in the text (lines 159-168).

**Answer to RC2**

**Summary:**

This paper aims to examine the relative impacts of the CFC chemistry and greenhouse gas effects on the large-scale circulation by the end of the century under a "world-avoided" type scenario. The authors demonstrate that both effects can contribute to varying degrees to changes in large-scale modes of variability and surface temperature. Overall, I think it is well-written and provides some interesting insights into what the circulation could have looked like and the relative roles of the two CFC effects.

**Technical comments:**

1. For Figures 8b, c, d and F2, please label the panels according to the season. Also for Figure 8, the legend text is not showing up for me (on the screen and when I print on paper).

   Thank you for pointing out the missing labels and legend. We contacted the editorial support about it as they appear in our original document. Unfortunately, they cannot adapt the pre-print document.

2. For Figure 8a, can you add the total MPA temperature time series to this plot?

   For the period 1980-2020, we only observed little temperature change (see also Fig. 4 in Egorova et al. (2023)) and therefore we decided to only show the temperature evolution from 2020 on. This way we are able to zoom in on the years around 2050 when the temperature curves start to diverge. We clarified this in the text (lines 277-279, 289-292).

3. In line 123, it is stated that "PSCs Type 2 ... are only allowed down to 50deg in the NH and SH in SOCOL." Can you clarify this? Are you saying that this parameterization only extends from zero to 50 deg latitude in each hemisphere?

   Exactly, SOCOL's PSC2 parametrization extends from 0–50° in each hemisphere. We clarified this in the text (see lines 125-126) and added a recent study from our group where PSCs in SOCOLv3.1 (the predecessor of SOCOLv4) were investigated (Steiner et al., 2021).

**Additional corrections**

**line 14:** We made the wording more precise that we are referring there to lower stratospheric and tropospheric winds.

**line 44:** We corrected "CO2" to $CO_2$.

**line 54:** We changed to wording from "prescribed atmospheric chemistry effects" to "prescribed chemistry"

**line 192:** We swapped the beginning and the end of the sentence from "Compared to the CFC chemical effect... for the CFC radiative effect." to "For the CFC radiative effect, ... compared to the CFC chemical effect."

**line 228:** We added Morgenstern et al. (2014) to our SAM discussion, since they observe a similar cancellation effects for direct and ozone-mediated effects when increasing GHG.

**line 167:** We replaced Ball et al. (2016) and Calvo et al. (2017) by better suited references (Haase et al., 2020; Waugh et al., 2009)

**lines 76, 272, 275:** We changed Egorova et al. (2022) to Egorova et al. (2023) since their pre-print is now accepted and published.

**lines 261-262:** We made the wording more precise.

**line 323:** We corrected the typo "largely shaped" to "largely shape"

**line 318:** We added Goyal et al. (2019).

**section titles 3.3, 3.4:** We added "tropospheric" in order to make clear that we are only refer to the tropospheric SAM and NAM.

**Figure 2:** We removed the white lines of the contour levels of panel a and b.

[revised manuscript text omitted]

---

## Author Response (AR2)

**Authors' Response to Review by Editor**

Franziska Zilker[1,4], Timofei Sukhodolov[2], Gabriel Chiodo[1], Marina Friedel[1], Tatiana Egorova[2], Eugene Rozanov[2,3], Jan Sedlacek[2], Svenja Seeber[1], and Thomas Peter[1]

[1]Institute for Atmospheric and Climate Science (IAC), ETH, Zurich, Switzerland
[2]Physikalisch-Meteorologisches Observatorium Davos/World Radiation Center, Davos, Switzerland
[3]St. Petersburg State University, St. Petersburg, Russia
[4]Swiss Federal Institute for Forest, Snow, and Landscape Research (WSL), Birmensdorf, Switzerland

**Correspondence:** Franziska Zilker (franziska.zilker@wsl.ch)

We thank the editor for his insightful comments and suggestions to our manuscript. We responded to them in detail below. The editor's comments are given in black and our answers are indicated in blue. We also explained minor corrections introduced by ourselves. The line numbers indicated in the response refer to the revised manuscript. The authors' track-changes file is attached with all changes highlighted (generated with latexdiff).

(i) Your abstract concludes with 'avoid regional amplifications of negative climate impacts' and your conclusions with 'regional amplification of adverse effects on surface climate'. The message is that the Montreal Protocol has avoided negative regional climate impacts. But as far as I can tell you don't really specify what these negative regional climate impacts might be – you simply point to changes the circulation. Can you spell out very briefly what the adverse effects might be – e.g. perhaps you are thinking that poleward contraction of the SH eddy driven jet in austral summer will reduce rainfall in southern Australia? Perhaps you can cite a paper that links the type of circulation changes that you are considering to specific 'adverse effects'?

Thank you for making us aware of missing examples. We added an example for the SH referring to Fig. 7b in Egorova et al. (2023), where we show that the more positive SAM phase would have shifted regional precipitation patterns and how it is similar to local climate change tendencies (Lee et al., 2023) (lines 245-251). For the NH, we added the increased precipitation in the Pacific sector again referring to Egorova et al. (2023) and Lee et al. (2023) (lines 271-274). We also put the regional warming in the Arctic region into perspective, by comparing it with the IPCC AR6 (Lee et al., 2023) (lines 299-302). We summarized the amplification of adverse effects in the conclusions (lines 350, 355-357).

(ii) I've seen a couple of typos – e.g. 'topical' in l236 should be 'tropical', the 'J.W.' in the cited reference in l285 should be removed. Please check for others.

Thank you. We have corrected the indicated typos and also checked for others.

**Additional corrections**

- "noMPA_noCFCRad" and "noMPA_CFCRadOff" were used interchangeably in the text. We decided to use "noMPA_CFCRadOff".

- We indicated in the figure captions for colormaps showing differences that the color saturation is different for negative and positive values (where applicable).

l. 10: We changed "vortex" to "wind" to avoid the misleading statement of a summertime polar vortex.

l. 115, 130 The preprint Egorova et al. (2022) was still cited which we changed to the publication Egorova et al. (2023)

l. 123 We changed ";" to "and" to allow for better readability of the citations in the text.

Figs 1, 8, D1: We corrected the legends from "noMPA_noCFCRad" to "noMPA_CFCRadOff" to be consistent with the rest of the text.

Figs. 7, 8: We moved them in the text, therefore the track-change file shows the deletion of the captions where they were before.

App. B: Appendix B was move to D to match its occurrence in the text.

**References**

Egorova, T., Sedlacek, J., Sukhodolov, T., Karagodin-Doyennel, A., Zilker, F., and Rozanov, E.: Montreal Protocol's impact on the ozone layer and climate, Atmospheric Chemistry and Physics Discussions, 2022, 1–19, https://doi.org/10.5194/acp-2022-730, 2022.

Egorova, T., Sedlacek, J., Sukhodolov, T., Karagodin-Doyennel, A., Zilker, F., and Rozanov, E.: Montreal Protocol's impact on the ozone layer and climate, Atmospheric Chemistry and Physics, 23, 5135–5147, https://doi.org/10.5194/acp-23-5135-2023, 2023.

Lee, J.-Y., J. Marotzke, G. Bala, L. Cao, S. Corti, J.P. Dunne, F. Engelbrecht, E. Fischer, J.C. Fyfe, C. Jones, A. Maycock, J. Mutemi, O. Ndiaye, S. Panickal, and T. Zhou: Future Global Climate: Scenario-based Projections and Near-term Information, in: Climate Change 2021 – The Physical Science Basis: Working Group I Contribution to the Sixth Assessment Report of the Intergovernmental Panel on Climate Change, edited by Masson-Delmotte, V., P. Zhai, A. Pirani, S.L. Connors, C. Péan, S. Berger, N. Caud, Y. Chen, L. Goldfarb, M.I. Gomis, M. Huang, K. Leitzell, E. Lonnoy, J.B.R. Matthews, T.K. Maycock, T. Waterfield, O. Yelekçi, R. Yu, and B. Zhou, pp. 553–672, Cambridge University Press, Cambridge, United Kingdom and New York, NY, USA, 1 edn., https://doi.org/10.1017/9781009157896, 2023.